# Self assembling nanoparticle enzyme clusters provide access to substrate channeling in multienzymatic cascades

Joyce C. Breger[1,8], James N. Vranish[1,2,8], Eunkeu Oh [3], Michael H. Stewart[3], Kimihiro Susumu[3], Guillermo Lasarte-Aragonés [1,4], Gregory A. Ellis[1], Scott A. Walper[1], Sebastián A. Díaz [1], Shelby L. Hooe[1,5], William P. Klein[1,5], Meghna Thakur[1,4], Mario G. Ancona[6,7] & Igor L. Medintz [1] ✉

Access to efficient enzymatic channeling is desired for improving all manner of designer biocatalysis. We demonstrate that enzymes constituting a multistep cascade can self-assemble with nanoparticle scaffolds into nanoclusters that access substrate channeling and improve catalytic flux by orders of magnitude. Utilizing saccharification and glycolytic enzymes with quantum dots (QDs) as a model system, nanoclustered-cascades incorporating from 4 to 10 enzymatic steps are prototyped. Along with confirming channeling using classical experiments, its efficiency is enhanced several fold more by optimizing enzymatic stoichiometry with numerical simulations, switching from spherical QDs to 2-D planar nanoplatelets, and by ordering the enzyme assembly. Detailed analyses characterize assembly formation and clarify structure-function properties. For extended cascades with unfavorable kinetics, channeled activity is maintained by splitting at a critical step, purifying end-product from the upstream sub-cascade, and feeding it as a concentrated substrate to the downstream sub-cascade. Generalized applicability is verified by extending to assemblies incorporating other hard and soft nanoparticles. Such self-assembled biocatalytic nanoclusters offer many benefits towards enabling minimalist cell-free synthetic biology.

The promise of synthetic biology (synbio) for biomanufacturing is touted as the next industrial revolution[1-5]. To date, development has primarily focused on creating optimized cell-based chassis to host designer heterologous enzymatic cascades yielding desirable products ranging from industrial precursors to fine specialty chemicals[6]. Along with removing competing metabolic pathways, researchers also seek to exploit naturally occurring phenomena that augment biosynthesis including confining reactions within intracellular compartments and accessing substrate channeling via enzymatic clustering, which is believed to allow cells to control selected metabolic processes such as glycolytic flux[6-13]. Such clustered transient multi-protein complexes consist of sequential enzymes that engage in substrate channeling and are termed metabolons[14-17]. Here, the enzymes are positioned close enough (0.1–1.0 nm) that the intermediary product of each enzyme

[1]Center for Bio/Molecular Science and Engineering, Code 6900, U.S. Naval Research Laboratory, Washington, D.C. 20375, USA. [2]Department of Chemistry, Engineering, and Physics, Franciscan University of Steubenville, Steubenville, OH 43952, USA. [3]Optical Sciences Division, Code 5611, U.S. Naval Research Laboratory, Washington, D.C. 20375, USA. [4]College of Science, George Mason University, Fairfax, VA 22030, USA. [5]National Research Council, Washington, D.C. 20001, USA. [6]Electronic Science and Technology Division, Code 6800, U.S. Naval Research Laboratory, Washington, D.C. 20375, USA. [7]Department of Electrical and Computer Engineering, Florida State University, Tallahassee, FL 32310, USA. [8]These authors contributed equally: Joyce C. Breger, James N. Vranish. ✉e-mail: igor.medintz@nrl.navy.mil

has a high probability of immediately being taken up as substrate by the next enzyme rather than diffusing away[18]. Exploiting this concept, Dueber designed a modular protein scaffold hosting three mevalonate biosynthetic enzymes with differing stoichiometry to match catalytic rates and increased product titer 77-fold in *E. coli*[19]. Engineering enzymatic clusters in vivo is non-trivial[20], and even fusing enzymes together is often insufficient since most intermediary diffusion rates are orders of magnitude faster than the corresponding enzymatic rates[21–24]. Wingreen provided one potential solution in the form of functional co-clusters where multiple copies of upstream and downstream enzymes are assembled into agglomerates[25]. Here, the large number of downstream enzymes in close proximity present significantly increased the probability of an intermediary from the first enzyme being utilized. They further showed control over a metabolic branch point between pyrimidine and arginine biosynthesis by directing synthetic flux through one arm of the branch. Their agglomerates emphasize enzyme density over internal cluster organization for achieving high kinetic flux, however, the question of which factor is more important remains a continuing source of debate in this area[11,26–29].

Although undeniably powerful, cell-based synbio approaches come with many functional limitations that significantly constrain their application space including: intolerance to non-natural substrates and toxic intermediates; inability to insert post-translationally modified eukaryotic enzymes into prokaryotic expression lines without complex re-engineering; the need for cellular viability; and production inefficiencies from competing pathways coupled to the wide range of turnover rates present across diverse enzyme families[30–32]. Cell-free systems partially address some of these issues, but many remain given that most cell-free systems are cellular extracts or reconstituted components thereof[33]. These limitations have motivated pursuit of minimalistic cell-free synbio systems consisting of just enzymes, substrate, and the cofactors required for a given cascade. Functional liabilities now become that of long-term enzymatic stability and inefficiency due to reaction-diffusion limitations and researchers have tried to address these, for example, by attaching enzymes in sequential order in close proximity on scaffolds in an effort to access channeled biocatalysis[29,34–37]. The potential of such systems for efficient, targeted biosynthesis using mix and match heterologous enzymes has driven concerted investment with both organic and inorganic scaffolding materials being prototyped including molecular organic frameworks, proteins, peptides, DNA, liposomes, cells, nanoparticles (NPs), and polymers[34,35,37–44]. These efforts have not been without controversy surrounding the choice of model enzyme systems used, how the enzymes are coassembled and attached to a scaffold, debate over the origins of any enhanced turnover observed in these assemblies, and whether channeling is actually even being achieved[22,24,45,46]. Wheeldon and Hess provided strong evidence that the ubiquitous coupled-glucose oxidase (GOx)/horse radish peroxidase model system does not engage in channeling especially when attached to DNA contrary to initial reports[23,27,45,47–50]. Rather, the observed catalytic enhancements were attributed to localized substrate-sequestration effects arising from the highly-localized DNA charge density and this conclusion has been reinforced by a recent study showing no enhancement when an unrelated three-enzyme system assembled in close proximity on DNA origami triangles[51].

Our prior efforts looked to understand the catalytic enhancements (e.g., 2–50× improvements in $k_{cat}$) observed in enzymes attached to NPs such as semiconductor quantum dots (QDs) or gold NPs (AuNPs)[52–54]. Work with amylase, glucokinase, phosphotriesterase, β-galactosidase, benzaldehyde lyase (Bal), alkaline phosphatase, and lactate dehydrogenase (LDH) revealed that enhancements arose from localized nanoenvironment and interfacial effects at the NP-solvent boundary, which served to alleviate key rate-limiting steps (e.g., enzyme-product release or $k_{off}$) and also from stabilizing enzyme

structures[55–61]. Moreover, enhancements were often dependent on NP size, with smaller diameter materials generally favoring catalytic increases[52]. When pyruvate kinase (PykA) was coassembled with LDH on QDs in a coupled enzymatic format, each enzyme's tetrameric structure induced QD cross-linking into small NP-enzyme clusters, which exhibited >100-fold enhancements in coupled-catalytic flux over that of an equivalent freely-diffusing enzyme. Detailed experiments, along with simulations, confirmed intermediate channeling between the QD-colocalized enzymes in the clusters[55]. Moreover, when Bal was recently paired with an alcohol dehydrogenase in similarly cross-linked QD enzyme clusters, their coupled enzymatic flux increased 50% despite the two enzymes differing by >10⁴ in catalytic rate and by three orders of magnitude in their respective Michaelis constant, $K_M$[61].

In the present work, we exploit this same NP-enzyme clustered strategy in far more complex self-assembled NP-enzyme systems incorporating from 4 up to 10 enzymes, see Fig. 1. Utilizing oxidative glycolysis as a model system, we show that biocatalytic cascades can self-assemble with NPs into nanoclustered aggregates that enable concerted substrate channeling and increase overall catalytic flux by orders of magnitude over that of freely-diffusing enzymes, the latter of which encounter significant diffusion limitations. We confirm the presence of channeling using classical experiments and further show that channeling efficiency is enhanced several-fold more by optimizing enzymatic stoichiometry with numerical simulations, switching from spherical QDs to 2D planar nanoplatelets, and ordering the NP-enzyme assembly process. Detailed analyses also characterize assembly formation and clarify structure-function properties. Further, a two-module approach is implemented to link cascades with overall unfavorable kinetics, while assemblies incorporating other hard and soft NPs show similar results and confirm the generality of the approach.

## Results

### Enzymes, nanoparticles, nanocluster formation, and assays

Details on materials, experimental formats, and analysis can be found in the subsequent Methods Section and Supplementary Information that accompanies this paper. The self-assembled biocatalytic NP-enzyme clusters depicted in Fig. 1 consist of enzymes defining a multistep cascade added to NPs in stoichiometric amounts with enzymes typically in a cumulative excess. Enzymes were all expressed with a terminal hexahistidine (His₆) for purification over Ni²⁺-nitrilotriacetic acid (NTA) resin, which also allows them to self-assemble to the ZnS shell of the different NP materials used here *via* metal-affinity coordination, see Fig. 1a[62,63]. This high-affinity, cooperative interaction ($K_d$ ~1 nM) occurs almost spontaneously and has been repeatedly verified as a robust QD bioconjugation strategy[62,64,65]. When binding a single type of enzyme-to-QDs, and not considering steric hindrance or cross-linking, self-assembly follows a Poisson distribution providing control over the average number of proteins assembled per QD simply through stoichiometry[62,64]. Analogous to what was observed with PykA and LDH assembly to QDs[55], as shown in Fig. 1b, NP clustering again occurs because many of the enzymes utilized are obligate dimers or tetramers (Table 1) and their multiple pendant-His₆ attach to multiple QDs bridging them into nanoclustered aggregates (vide infra).

The enzymatic pathways utilized are shown in Figs. 1c, 2. Along with their abbreviations, descriptive information about each enzyme is found in Table 1. The initial cascade consisted of the seven enzymes (7E) drawn from oxidative glycolysis that convert glucose into 3-phosphoglycerate (3-PG) while regenerating 2 ATP equivalents with NAD⁺ as a cofactor. Although NAD⁺ conversion to NADH by GPD is used to monitor this cascade's activity, and GPD is considered to separate upper and lower glycolysis, we include PGK, the enzymatic step following GPD, in this cascade. PGK's strongly-negative free energy ($\Delta_r G'^m = -20.2$ kJ/mol, see Supplementary Information Thermodynamic analysis and ΔG) helps drive flux forward, whereas that of

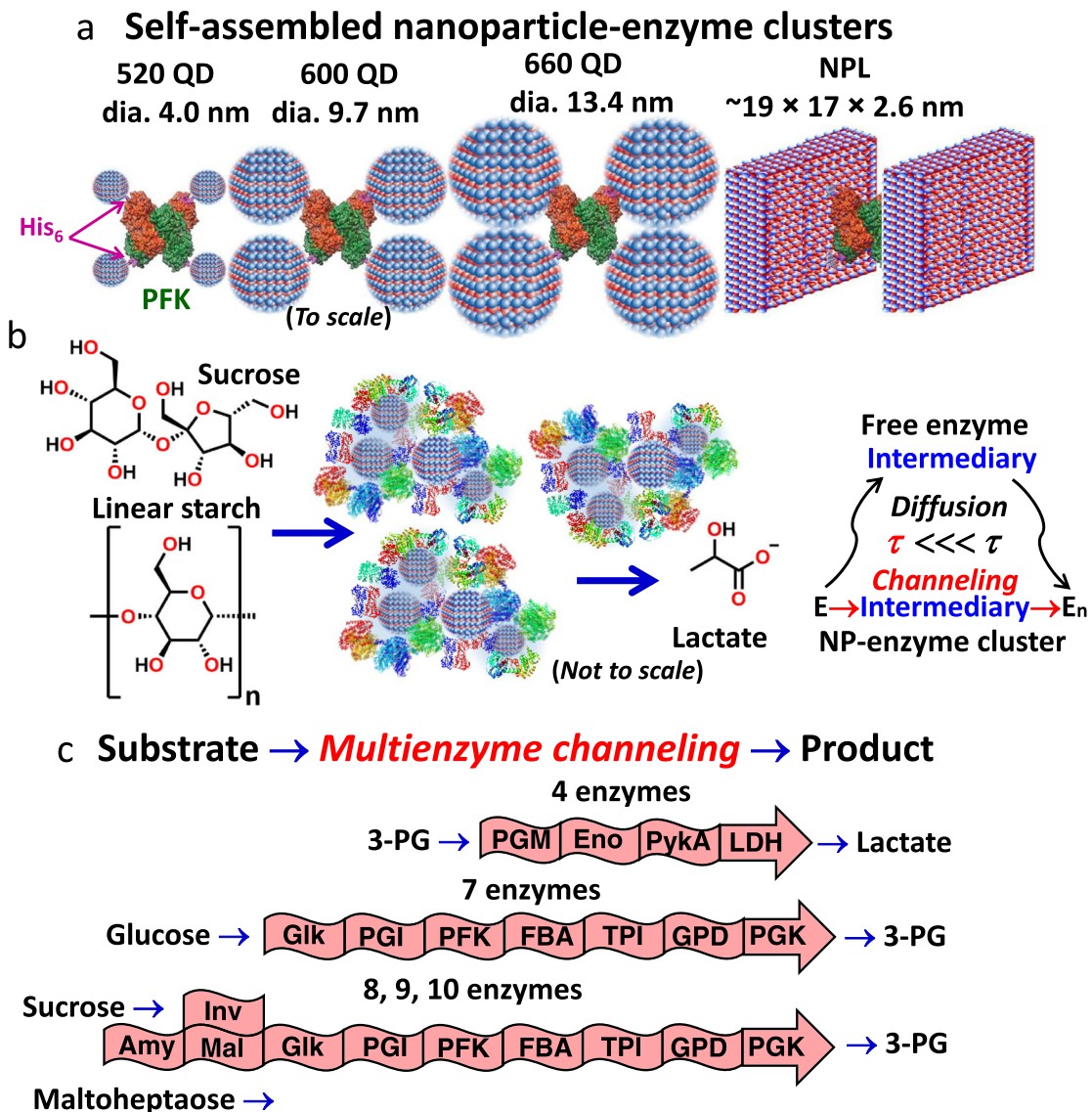

## a Self-assembled nanoparticle-enzyme clusters

**520 QD** dia. 4.0 nm  **600 QD** dia. 9.7 nm  **660 QD** dia. 13.4 nm  **NPL** ~19 × 17 × 2.6 nm

His₆  PFK  (*To scale*)

## b

Sucrose

Linear starch

Lactate

(*Not to scale*)

Free enzyme
**Intermediary**
*Diffusion*
$\tau <<< \tau$
*Channeling*
E→**Intermediary**→Eₙ
**NP-enzyme cluster**

## c Substrate → *Multienzyme channeling* → Product

**4 enzymes**
3-PG → | PGM | Eno | PykA | LDH | → Lactate

**7 enzymes**
Glucose → | Glk | PGI | PFK | FBA | TPI | GPD | PGK | → 3-PG

**8, 9, 10 enzymes**
Sucrose → | Inv |
| Amy | Mal | Glk | PGI | PFK | FBA | TPI | GPD | PGK | → 3-PG
**Maltoheptaose →**

**Fig. 1 | Nanoparticles, catalytic nanoclusters, and enzymatic pathways.**
**a** Multiple His₆-termini (purple) on the multimeric enzymes coordinate to the NP surfaces and functionally crosslink them into nanoclusters as shown with phosphofructokinase I (PFK, PDB #1PFK) and the three different sizes of QDs along with NPLs used at a scale relative to PFK. NPLs are shown angled for perspective.
**b** Schematic depicting the self-assembled QD enzyme clusters forming multi-enzyme cascades that are the focus of this study. QDs are mixed with stoichiometric ratios of enzymes that constitute a targeted cascade and self-assemble into nanoclusters. The addition of initial substrates, such as linear starch, is then processed into the product by the multienzyme cascade in the cluster, which exploits localized intermediary channeling. Forming into NP-enzyme clusters and engaging in multistep channeling increases the overall catalytic flux by orders of magnitude over that of freely-diffusing enzymes, which encounter significant diffusion limitations. The former substantially reduces the overall transient time ($\tau$) for that reaction. **c** Examples of the multienzyme cascades assembled into nanoclusters and explored here. See Table 1 for full enzyme names.

GPD (15.9 kJ/mol) favors the reverse gluconeogenic reaction[66,67]. The 7E cascade was subsequently expanded to 8, 9, and 10 enzymes by adding upstream saccharification steps processing maltoheptaose into glucose or hydrolyzing sucrose. A four-enzyme (4E) cascade converting 3-PG into lactate was also investigated (Fig. 1c). GOx is utilized as a branch point competitor for probative purposes (Fig. 2). These enzymes were chosen because they are connected within the known glycolytic metabolon[14–17], their cascaded function is well characterized, and extensive structural data about them is available. For NPs, 520, 600, and 660 nm emitting CdSe/CdS/ZnS core/shell/shell QDs with average diameters of ~4.0, 9.7, and 13.4 nm, respectively, were utilized[68]. 585 nm emitting CdSe/ZnS core/shell nanoplatelets (NPLs, four monolayers CdSe) with an average LWH of ~19.2 × 17.3 × 2.6 nm were also used to exploit their extended flat surface areas (Fig. 3a)[69]. Following synthesis, native NP hydrophobic growth ligands were cap-

exchanged with the zwitterionic compact ligand CL4 to provide colloidal stability in the buffer (structure in Supplementary Fig. 1)[70]. Enzyme ability, individually and jointly, to assemble on NPs was confirmed by agarose gel mobility shift assays (Supplementary Figs. 14–33). Each enzyme's catalytic activity when free in solution or when NP-assembled was confirmed with mass-spectral analysis (Supplementary Table 35). All NPs were also tested to verify that they did not catalyze any reaction steps themselves since metallic nanoparticulates are commonly catalytically active[71].

Channeling is a nanoscale phenomenon that, while often misconstrued as improving an enzyme's activity or rate, refers to any mechanism that limits the out-diffusion of a reaction intermediate and thereby increases the probability that it will instead encounter the next enzyme in the cascade. Therefore, channeling will be observable in the diffusion-limited regime whenever the catalytic rate is much greater

**Table 1 | Enzymes utilized and some of their relevant properties**

| Enzyme (Abbrev.) | EC # | Active structure | Monomer[a] ~MW (kD) | PDB #[b] | [d]#/QD 4 nm | 9.7 nm | 13.4 nm | NPL | Net charge pH 7.5 / 8.5 | Ref. |
|---|---|---|---|---|---|---|---|---|---|---|
| Amylase (Amy) | 3.2.1.1 | Monomer | 59.6 | 1B90 | 4–10 | 18–25 | 32–39 | 24–36 | −4.4 / −9.3 | [130] |
| Maltase (Mlt) | 3.2.1.20 | Monomer | 74.5 | 3A47 | 4–8 | 16–21 | 26–34 | 21–31 | −14.4 / −21.2 | [131] |
| Invertase (Inv) | 3.2.1.26 | Tetramer | 47.2 | 1H6D | 4–6 | 11–17 | 18–29 | 14–24 | −21.2 / −24.5 | [127] |
| Glucokinase (Glk) | 2.7.1.1 | Dimer | 36.9 | 1Q18 | 6–7 | 19–29 | 34–43 | 25–33 | −4.1 / −9.3 | [132] |
| Phosphoglucose isomerase (PGI) | 5.3.1.9 | Dimer | 63.7 | 3NBU | 4–7 | 14–20 | 22–34 | 17–23 | −9.3 / −14.1 | [133] |
| Phosphofructokinase I (PFK) | 2.7.1.11 | Tetramer | 34.6 | 1PFK | 5–9 | 13–21 | 22–36 | 17–24 | −6.5 / −10.2 | [134] |
| Fructose-bisphosphate aldolase (FBA) | 4.1.2.13 | Dimer | 41.3 | 1B57 | 5–9 | 18–27 | 29–45 | 23–30 | −9.9 / −14.7 | [135] |
| Triose phosphate isomerase (TPI) | 5.3.1.1 | Dimer | 29.1 | 1TMH | 7–12 | 23–32 | 37–43 | 29–38 | −5.2 / −8.8 | [136] |
| Glyceraldehyde-3-phosphate dehydrogenase (GPD) | 1.2.1.12 | Tetramer | 37.7 | 1DC3 | 4–6 | 13–20 | 20–34 | 16–28 | −0.6 / −3.2 | [137] |
| Phosphoglycerate kinase (PGK) | 2.7.2.3 | Monomer | 43.3 | 1ZMR | 5–12 | 19–28 | 31–43 | 29–36 | −9.6 / −13.3 | [138] |
| Phosphoglycerate mutase (PGM) | 5.4.2.11 | Monomer | 58.3 | [c]1EJJ | 4–10 | 15–23 | 25–36 | 24–37 | −17.7 / −22.0 | [139] |
| Enolase (Eno) | 4.2.1.11 | Dimer | 47.8 | 1E9I | 5–11 | 16–25 | 26–41 | 21–33 | −9.2 / −12.6 | [140] |
| Pyruvate kinase II (PykA) | 2.7.1.40 | Tetramer | 53.5 | 2E28 | 3–9 | 10–19 | 16–27 | 13–26 | −1.4 / −5.7 | [141] |
| Lactate dehydrogenase (LDH) | 1.1.1.28 | Tetramer | 38.7 | 3WX0 | 4–10 | 12–27 | 20–33 | 16–32 | −8.6 / −13.5 | [142] |

EC – Enzyme Commission.
The genes encoding each enzyme were cloned from *E. coli* except amylase (*B. cereus*), maltase (*S. pombe*), and invertase (*Zymomonas mobilis*).
[a]Determined using http://protcalc.sourceforge.net/cgi-bin/protcalc from the cloned protein sequence.
[b]Structures used to derive fitting on QDs.
[c]Closest homolog.
[d]Estimated range of ratio of each enzyme that can assemble onto diameter = 4 nm (520 QD, 4.0 ± 0.4 nm), 9.7 nm (600 QD, 9.7 ± 1.0 nm), 13.4 nm (660 QD, 13.4 ± 1.3 nm) QD and 17 × 19 × 3 nm NPLs. Determined as in ref. [55,65] while accounting for size and binding orientation as described in the Supplementary Information.

than the diffusion rate. Since the latter is proportional to:

$$\min\left([E]^{2/3}, [I]^{2/3}\right) D \tag{1}$$

where $D$ is the diffusion constant (units of distance²/time); the potential for seeing channeling exists whenever enzyme concentrations $[E]$ are low and at early times before intermediate concentrations $[I]$ buildup[11,27,29,72]. Given this, we focus experimentally on assay activity at early times with low enzyme concentrations. Diffusional modeling and simulations suggested this required low nM enzyme concentrations[7,25,27,29,73]. NPs preassembled with enzymes, were thus serially diluted to final reaction concentrations of 5, 2.5, 1.25, and 0.63 nM NP with substrate and cofactors present in vast excess to meet Briggs–Haldane conditions, except for NAD⁺/NADH which, although still saturating, was adjusted to match the plate reader's dynamic measurement range[21]. Data presented typically use one of the dilutions for representative examples and these are indicated where appropriate. Enzyme presence in an assay is defined as its ratio per individual NP present in that assay (typically ranging from 1:1 up to a larger multiple) unless stated otherwise. As expected, increasing enzyme concentrations above those corresponding to these low nM concentrations moved the reactions out of the diffusion-limited regime where channeling effects are manifested, see Supplementary Figs. 47, 48 (vide infra)[72]. Kinetic descriptors for each enzyme, both on/off NP include maximum velocity ($V_{max}$), the Michaelis constant ($K_M$), catalytic rate ($k_{cat}$), enzyme efficiency ($k_{cat}/K_M$), and specific activity (SA) are found in Table 2[21]. Kinetic values are termed "apparent" since it is uncertain if enzyme activity when it is displayed on a NP satisfies all classical Michaelis–Menten (MM) assumptions; nevertheless, they still serve as excellent comparators of activity in the different configurations[37,43,53,54,74].

**Initial assays, optimization of enzyme ratios, and assay conditions**

Experiments initially utilized the 7E system (Fig. 2) with 520 QDs and enzyme/QD ratios chosen empirically (Supplementary Table 2) based on their relative catalytic activity and the estimated QD display

capacity for each enzyme (Table 1). In comparison to freely-diffusing enzymes, a substantial increase in the cascade's catalytic flux was observed when the enzymes were self-assembled with QDs, e.g., compare the blue (Empirical free) and red (Empirical on QD) progress curves in Fig. 3b where NADH turnover increased ~30× from 10 to >300 μM. Cognizant that optimizing the relative enzyme ratios could further improve overall catalytic flux by addressing mismatches in turnover rates, detailed kinetic simulations were undertaken as described in Supplementary Information Kinetic simulations. This required exhaustive characterization of each enzyme's kinetic profile when free and NP-attached (Table 2). These experiments showed that individual turnover rates for 7 of the 14 enzymes were sped up when NP-attached, manifest across a range of magnitudes, varied with enzyme/NP ratio, and generally were largest at the lowest ratio, similar to previous observations (see also Supplementary Tables 14–34)[52,55–57,59,75]. Two rounds of numerical optimization were undertaken, and based on the results, assays were performed and compared with the empirical ratio results (Fig. 3b Opt 2–pink, Supplementary Table 2). Optimized ratios were found to increase NADH production by ~55× to 550 μM. Additionally, Opt 2 reactions plateaued in less time due to increased initial flux (compare slopes of red *vs.* pink curves in Fig. 3b) as expected for intermediate channeling since its contribution should be most significant in the reaction's initial stages[27,29].

Contributions of NP dimensionality and shape on cascade flux were next evaluated using the three different QD sizes and NPLs as assembly scaffolds (Fig. 1a). Estimated surface areas for these materials range from 50 up to 564 nm² for the 520 versus 660 QDs and ~854 nm² for the NPLs while the surface to volume (S/V) ratios range from 1.5/nm down to 0.5/nm for the QDs and around 1/nm for the NPLs. Identical concentrations of the four NP samples were assembled with the same ratios of enzymes (Opt 2, Fig. 3c), where the smaller 520 QDs outperformed the other QD materials, while NPLs were far superior to the spherical NPs. This aligns with our previous data indicating that smaller NPs outperform larger size NPs and portends that nanoscale material will outperform larger, micro/macro-scale particles[52,76]; clearly, however, surface area and S/V alone are not the critical determinants for

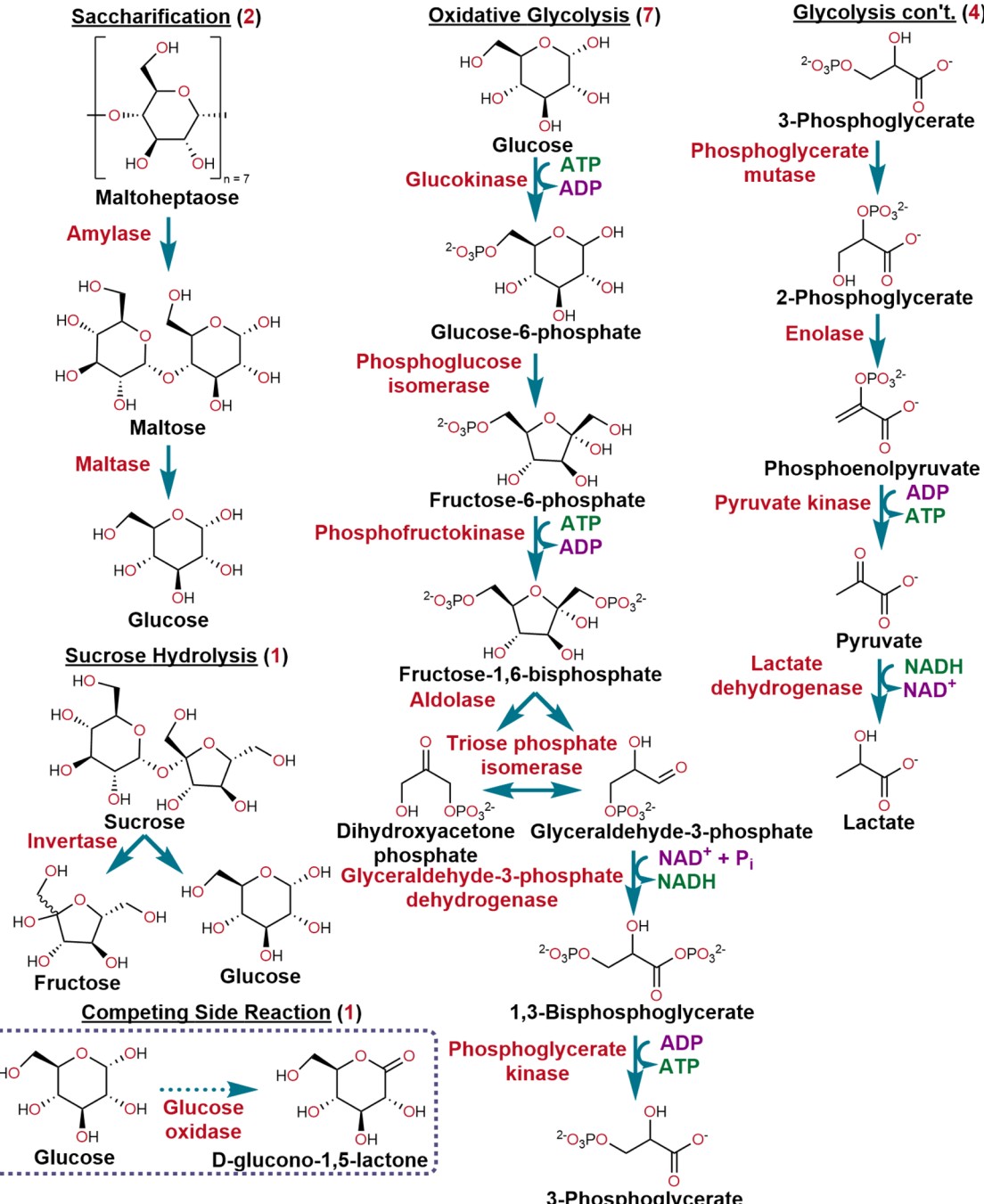

**Fig. 2 | Enzyme pathways utilized in this study.** The primary 7 enzyme (7E) pathway processes glucose to 3-phosphoglycerate (3-PG) and this is extended by adding upstream saccharification of amylose or sucrose to glucose. The 4 enzymes processing 3-PG to yield lactate constitute a second downstream cascade. The cascades terminating in the production of 3-PG were assayed by monitoring NADH formation, while those utilizing 3-PG to produce lactate followed NADH consumption[55,56]. Glucose oxidase is used in a competitive reaction format to test for channeling. Enzymatic steps are indicated in blue, enzyme names in red,

substrate/intermediary/products in black, and cofactors in maroon and green. Genes for each enzyme were cloned or chemically assembled, and their sequences were confirmed by DNA sequencing. Enzymes were expressed in *E. coli*, purified, aliquoted, and snap-frozen in 25% glycerol for −80 °C storage. Fresh aliquots of each enzyme were utilized in experiments. Enzymatic activity was assayed using Tecan Spark microtiter plate readers (see the Methods and Supplementary Information). Assays are all performed in the dark.

improving flux in the composite nanoclusters. To determine optimal cascade conditions, the effects of varying temperature and pH were next surveyed for the 520 QDs and NPLs assembled with Opt 2 ratios. Increasing assay temperature from 18 to 30 °C dramatically increased catalytic flux; however, increases above 35 °C were detrimental, presumably due to enzyme denaturation (Fig. 3d). Catalytic throughput was found to be significantly impaired below pH 6 but increased

significantly at pH 8.2 and then decreased slightly for the QDs and free enzyme control at pH 11.4 (Fig. 3e); HEPES buffer pH 8.2 and 30 °C were used for subsequent assays. Subsequent experiments utilized either NPLs or 520 QDs somewhat interchangeably as they manifested the largest increases in catalytic flux.

Another set of experiments looked at increasing NP concentrations at a fixed enzyme concentration using Opt 2 ratios. For QDs

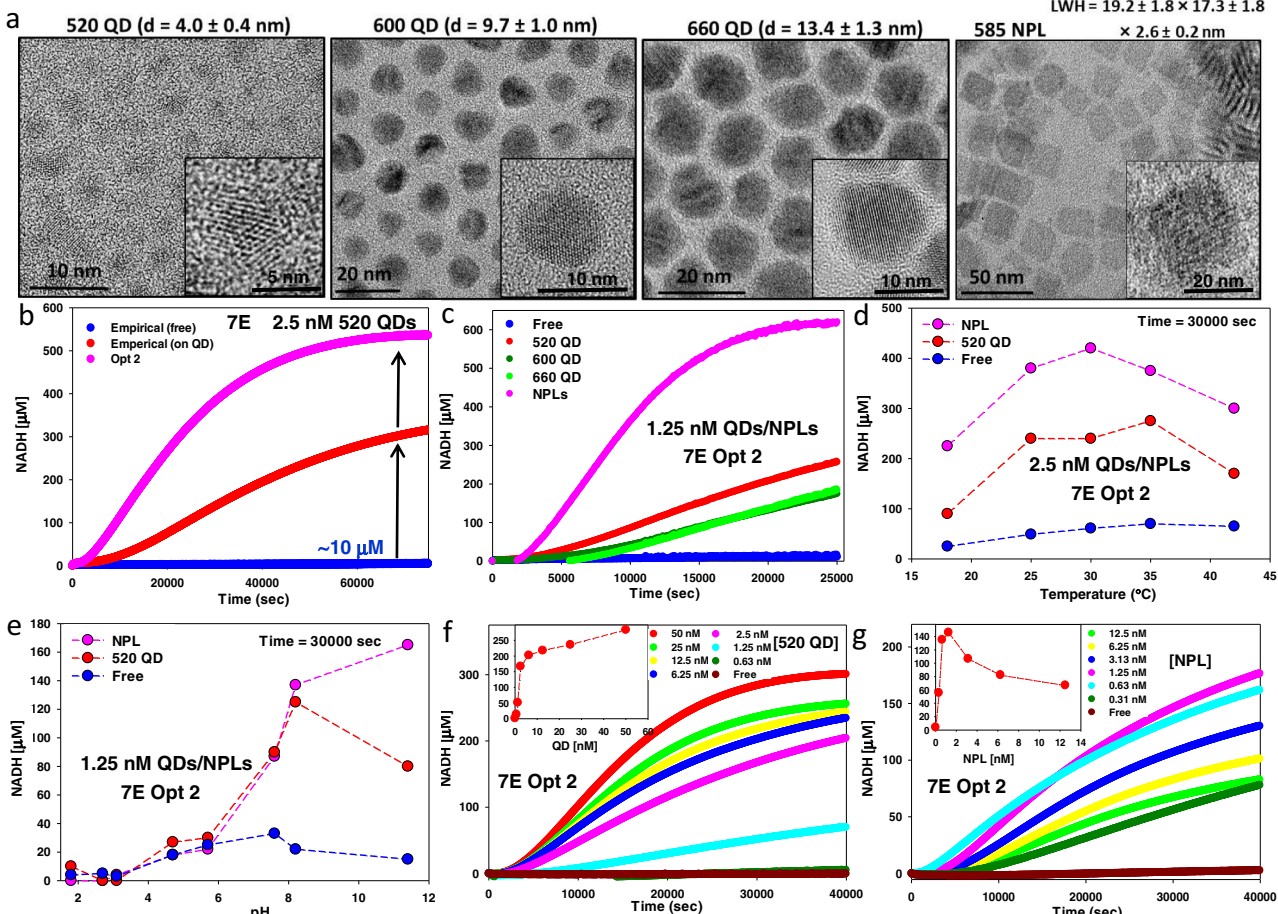

**Fig. 3 | Nanoparticles and catalytic performance in the 7 enzyme glucose→3-PG cascade. a** TEM micrographs of the QD and NPL materials utilized along with their average diameter (d) or size. Representative high-resolution micrograph shown inset. LWH = length × width × height. The terms QD and NPL distinguish each material, while NP refers to them collectively. **b** Representative progress curve measuring NADH conversion over time for 520 QD clusters assembled with the 7E system (glucose→3-PG) at empirical enzyme ratios (red) vs. same enzyme free in solution (blue). 10 μM indicates the final amount of NADH converted in the free enzyme assay. Progress curves assembled using optimized enzyme ratios per QD determined after two consecutive rounds of numerical simulation (pink - Opt 2). Free enzyme controls for optimization had identical results as that of the empirical sample. 520 QD concentration = 2.5 nM. **c** Progress curve comparing NADH conversion using Opt 2 enzyme ratios with the three different sized QD samples, NPLs, and free enzyme. QD/NPL = 1.25 nM. **d** NADH conversion at 30,000 s for 2.5 nm

QD/NPL assembled with Opt 2 enzyme ratios vs. reaction temperature. **e** NADH conversion at 30,000 s for 2.5 nm QD/NPL assembled with Opt 2 enzyme ratios vs. different pH (HEPES buffer, pH = 1.8, 2.7, 3.1, 4.7, 5.7. 7.6. 8.2, and 11.4). Comparison of NADH conversion over time for fixed enzyme concentrations at Opt 2 ratios vs. indicated increasing concentrations of QD (**f**) or NPL (**g**). Inset plots the relative amount of NADH converted vs. QD/NPL concentration present at 30,000 s. Starting NAD⁺ concentration was -1.13 mM for reactions. For each reaction shown in this Figure and those below, an individual plot averaged from the triplicate assays undertaken is shown for simplicity. Kinetic values in Table 2 are derived from triplicate assays and are listed along with their standard deviation. The data shown indicate the corresponding NP dilution. Enzyme concentrations are described as the ratio per NP; Supplementary Table 2 lists all pertinent enzyme ratios used. Assays were performed in at least triplicate and always included free enzyme controls with equivalent enzymes present without NPs.

(Fig. 3f), the observed bimodal response shown in the inset reveals that increasing the QD concentration -10× from 0.63 to 6.25 nM dramatically improved catalytic flux, while further increases up to 50 nM QD had a less intense rate increase. In contrast, the NPL samples (Fig. 3g) displayed significant catalytic increases from 0.31 nM up to 1.25 nM (-4× increase in concentration), with flux decreasing consistently from there to the highest NPL concentration. We hypothesize that at higher NPL concentrations, larger NPL enzyme aggregates formed, and these began to precipitate during the assays. These results suggest that there is an optimal enzyme/NP ratio for accessing proximity channeling in a given enzymatic configuration and that this optimum may depend on NP type. In examining the enzyme assay plots in Fig. 3 and those described below, optimized channeling sometimes increases the kinetic flux to the point that apparent saturation is reached in the assay's time window. It is important to note that in all the other experimental configurations plotted along with this type of data, the trajectories indicate a similar rise to saturation albeit at a much slower

rate as expected. We also do not account for any effects from the reverse gluconeogenic reactions, which will be initiated when a high enough concentration of a given intermediary is reached and which may slow down flux and final product formation.

## Analysis of nanoparticle-enzyme cluster formation

The above results motivated interest in better understanding NP-enzyme cluster assembly and how this contributes to improved catalysis. Detailed physicochemical characterization of the nanoclusters was undertaken, focusing mostly on the 7E system with Opt 2 ratios and the 520 QDs. Select dynamic light scattering (DLS), transmission electron microscopy (TEM), and FRET assays were applied to determine cluster presence and size along with the average number of QDs incorporated per cluster. Enzyme incorporation into the clusters was also probed with agarose and polyacrylamide gel electrophoresis (PAGE) mobility assays and FRET. These analyses were relatively consistent and confirmed not only cluster formation and enzyme

**Table 2 | Apparent kinetic values for each enzyme and as assembled on nanoparticles**

| Enzyme | Free | | | | | On QD[a] | | | | | Max on-NP enhancement[b] | |
|---|---|---|---|---|---|---|---|---|---|---|---|---|
| | $V_{max}$ ($\mu$M s$^{-1}$) | $K_M$ ($\mu$M) | $k_{cat}$ (s$^{-1}$) | $k_{cat}/K_M$ (mM$^{-1}$ s$^{-1}$) | S.A. ($\mu$mol min$^{-1}$ mg$^{-1}$) | $V_{max}$ ($\mu$M/s) | $K_M$ ($\mu$M) | $k_{cat}$ (s$^{-1}$) | $k_{cat}/K_M$ (mM$^{-1}$ s$^{-1}$) | S.A. ($\mu$mol min$^{-1}$ mg$^{-1}$) | QD | NPL |
| Amy | 0.063 ± 0.001 | 213 ± 15 | 25.0 ± 0.4 | 118 ± 8 | 25.2 ± 0.4 | 0.062 ± 0.004 | 212 ± 28 | 24.9 ± 1.9 | 118 ± 8 | 25.0 ± 2.0 | --- NS --- | --- NS --- |
| Mlt | 0.001 ± 0.001 | 335 ± 80 | 0.40 ± 0.03 | 1.20 ± 0.30 | 0.3 ± 0.0 | 0.046 ± 0.017 | 347 ± 90 | 15.3 ± 5.8 | 43.8 ± 12.4 | 12.2 ± 4.7 | ~50× $k_{cat}$ (0.25-0.5/QD) | ~125× $k_{cat}$ (0.25-1/NPL) |
| Inv | 0.014 ± 0.001 | 191 ± 100 | 0.07 ± 0.01 | 0.39 ± 0.21 | 0.024 ± 0.001 | 0.048 ± 0.005 | 2504 ± 954 | 0.26 ± 0.03 | 0.05 ± 0.02 | 0.082 ± 0.008 | --- NS --- | --- NS --- |
| Glk | 0.013 ± 0.001 | 36 ± 10 | 2.6 ± 0.1 | 73 ± 21 | 4.2 ± 0.2 | 0.154 ± 0.032 | 109 ± 14 | 30.7 ± 6.3 | 281 ± 44 | 54.1 ± 4.8 | ~12× $k_{cat}$ (0.2-1/QD) | ~7× $k_{cat}$ (0.25/NPL) |
| PGI | 0.261 ± 0.006 | 393 ± 31 | 2481 ± 53 | 6313 ± 518 | 2338 ± 50 | 0.214 ± 0.009 | 398 ± 46 | 2033 ± 89 | 5146 ± 472 | 1886 ± 86 | --- NS --- | --- NS --- |
| PFK | 0.030 ± 0.001 | 202 ± 21 | 10.1 ± 0.2 | 50.2 ± 5.3 | 17.6 ± 0.4 | 0.029 ± 0.002 | 203 ± 25 | 9.8 ± 0.7 | 48.6 ± 4.3 | 17.0 ± 1.3 | --- NS --- | ~4× $k_{cat}$ (0.25NPL) |
| FBA | 0.020 ± 0.001 | 216 ± 23 | 6.8 ± 0.2 | 31.4 ± 7.3 | 9.8 ± 0.2 | 0.017 ± 0.001 | 160 ± 20 | 5.7 ± 2.8 | 35.8 ± 4.3 | 8.2 ± 0.4 | --- NS --- | --- NS --- |
| TPI | 0.582 ± 0.035 | 1717 ± 236 | 194 ± 12 | 113 ± 17 | 401 ± 24 | 0.677 ± 0.116 | 2628 ± 693 | 225.8 ± 38.3 | 88 ± 10.5 | 466 ± 79 | --- NS --- | --- NS --- |
| GPD | 0.010 ± 0.001 | 3494 ± 1039 | 3.2 ± 0.04 | 0.92 ± 0.28 | 1.2 ± 0.1 | 0.028 ± 0.014 | 1790 ± 844 | 9.6 ± 4.8 | 5.4 ± 3.7 | 3.8 ± 1.9 | ~5× $k_{cat}$ (0.25/QD) | ~3× $k_{cat}$ (0.25/NPL) |
| PGK | 0.054 ± 0.003 | 1048 ± 171 | 18.1 ± 0.9 | 17.3 ± 3.0 | 25.1 ± 1.3 | 0.157 ± 0.013 | 933 ± 72 | 53.1 ± 4.6 | 56.9 ± 7.8 | 73.6 ± 6.3 | ~3× $k_{cat}/K_M$ (0.1/QD) | ~2.3× $k_{cat}/K_M$ (0.25/NPL) |
| PGM | 0.012 ± 0.001 | 3584 ± 1138 | 1.0 ± 0.1 | 0.285 ± 0.1 | 1.1 ± 0.1 | 0.063 ± 0.006 | 3344 ± 514 | 5.3 ± 0.5 | 1.6 ± 0.2 | 5.4 ± 0.5 | 6× $k_{cat}$ (2/QD) ~6× $k_{cat}/K_M$ (1/QD) | ~4× $k_{cat}/K_M$ (0.5/NPL) |
| Eno | 0.069 ± 0.001 | 2040 ± 72 | 22.9 ± 3.0 | 11.2 ± 4.3 | 28.7 ± 3.7 | 0.095 ± 0.017 | 2361 ± 249 | 31.1 ± 5.6 | 13.1 ± 1.3 | 39.0 ± 7.0 | --- NS --- | --- NS --- |
| PykA[c] | ---- | 690 ± 70 | 277 ± 9 | 401 ± 32 | 310 ± 10 | ---- | 920 ± 80 | 214 ± 6 | 232 ± 21 | 240 ± 7 | --- NS --- | --- NS --- |
| LDH | 0.24 ± 0.03 | 31470 ± 6901 | 95.5 ± 10.4 | 3.04 ± 0.74 | 37 ± 4 | 0.51 ± 0.03 | 11732 ± 3578 | 202 ± 7.6 | 19.1 ± 7.2 | 79.6 ± 4.0 | ~10× $k_{cat}/K_M$ (0.25/QD) | ~10× $k_{cat}/K_M$ (0.25/NPL) |

Values were estimated using standard Michaelis–Menten (MM) formalism.

All values are given per unit monomer of enzyme/SA given per unit of enzyme.

SA specific activity, NS none or not significant.

[a]On QD/NPL values averaged from data collected at a variety of enzyme per QD/NPL ratios, see Tables S13–S33 for specific values at each ratio.

[b]Fold enhancement for specified activity when QD or NPL displayed over equivalent concentration free enzyme (ratios where maximum enhancement observed and averaged from only the ratios showing enhanced results). PGK kinetics were collected from 3-phosphoglycerate conversion to 1,3-bisphosphoglycerate.

[c]Values from ref. [55] PGI kinetics collected from fructose-6-phosphate to glucose-6-phosphate conversion.

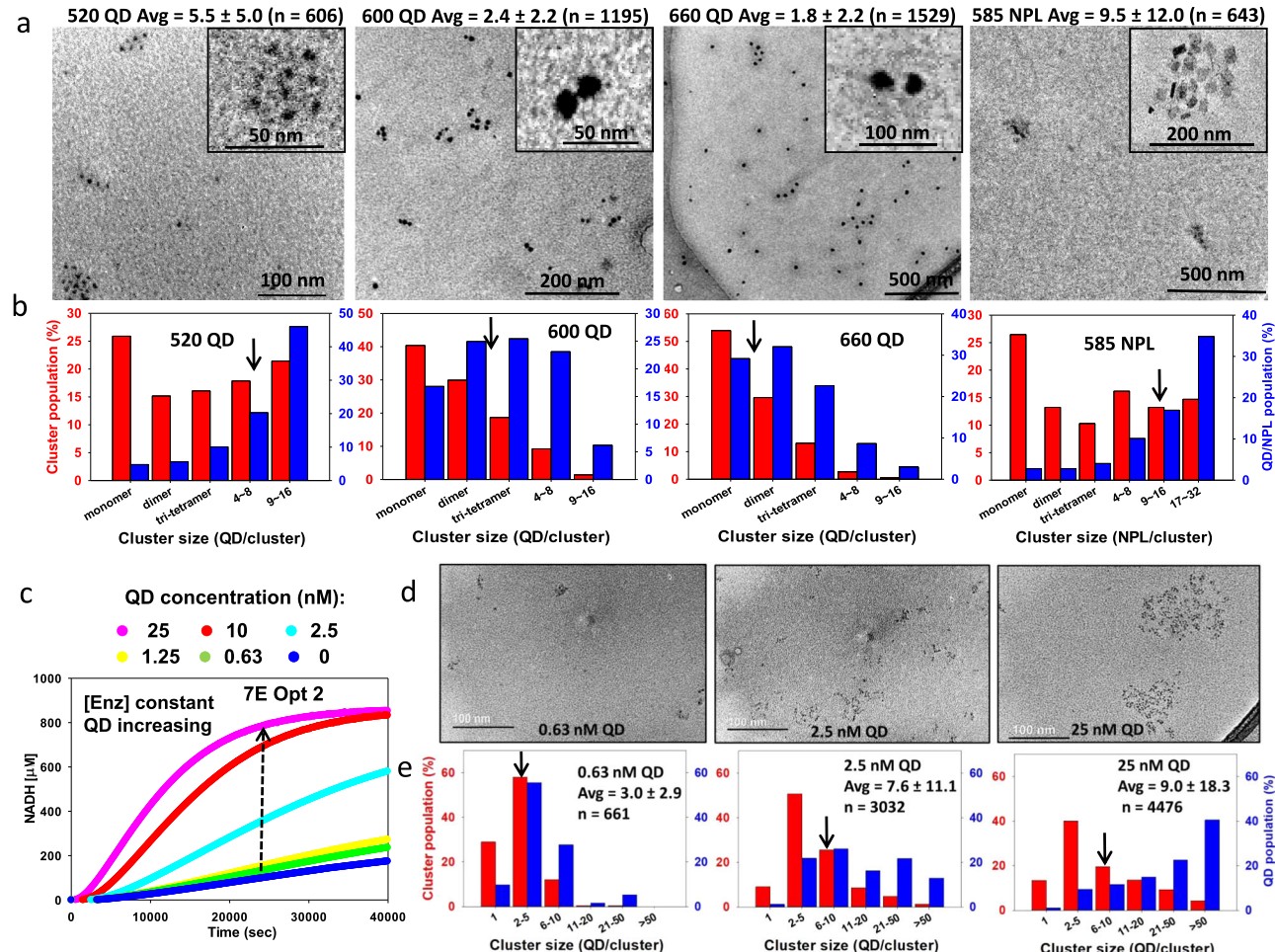

**Fig. 4 | TEM characterization of nanoparticle-enzyme clusters. a** Left to the right, representative TEM micrographs of clusters formed with 520 QDs, 600 QDs, 660 QDs, and 585 NPL materials using the 7E cascade at Opt 2 ratios with QD = 6.25 nM and NPL = 1.25 nM. The average cluster size is given above the micrograph, along with the number of QDs counted in that determination. Inset, representative high-resolution micrograph of an individual cluster. In interpreting these images, it should be remembered that changes may have occurred either in deposition on the TEM grids or in the high vacuum of the TEM. Enzymes can be seen in the TEM images as shading around some of the 520 and 660 QDs. **b** Corresponding bar plots for each sample in (**a**) showing the distribution of cluster sizes present (red) and the number of QDs per cluster size (blue). **c** Representative progress curves for assay data from 7E system Opt 2 ratios with enzyme concentration fixed (Glk 5.5, PGI 1, PFK 9, FBA 12, TPI 1, GPD 27, PGK 7.5 nM) as assembled with the indicated increasing concentrations of 520 QD. Representative TEM images (**d**) with corresponding cluster analysis bar plots (**e**) for the 0.63, 2.5, and 25 nM samples in panel **c**. The full cluster analysis is in Supplementary Figure 39. Black arrows in panels **b**, **e** show the approximate location of the average cluster size for that assembly. Three of the TEM experiments, including those shown here in **a**, **b**, **d**, and **e**, were replicated and returned essentially the same distributions as shown. All other data were collected from a single set of samples.

incorporation into the clusters, but also that multimeric enzymes were essential to forming clusters (confirming that the clusters result from enzyme cross-linking) and that cluster size could be controlled by varying the ratio of NP to total enzyme. See Supplementary Information Physicochemical analyses of NP-enzyme cluster formation for detailed discussion.

Results of the TEM analyses are summarized in Fig. 4a with images of the 520, 600, 660 QDs (6.25 nM), and NPLs (1.25 nM) assembled with identical 7E concentrations at Opt 2 ratios. Although these images were of material taken directly from the assays, one should be cautious in interpretation because of possible effects of TEM sample preparation and because of the limited number of samples examined; we thus qualify them as semi-quantitative. Nevertheless, they were of value for identifying changes in clustering distributions (*i.e.*, number of QDs or NPs per cluster and the frequency of given cluster sizes observed) for different NP/enzyme combinations and in seeing how these changes might correlate with enzyme activity. Regarding the 520 QDs (leftmost column in Fig. 4a, b), the clusters observed contain an average of ~5.5 QDs, with the QDs found predominantly in larger-sized clusters.

600 and 660 QD samples formed smaller average cluster sizes of 2.4 and 1.8, respectively, with most QDs found in smaller-size clusters. The NPL samples yielded a far different profile with larger cluster sizes averaging ~9.5 NPLs/cluster. Similar to the 520 QD sample profile, most NPLs were incorporated into larger clusters, which now include a larger 17–32 bin size. Despite their somewhat comparable surface areas and S/V ratios, the NPL's flat shape appears to be key to forming larger clusters. We note that cluster formation *via* enzyme cross-linking appears to be an example of the classical diffusion-limited aggregation (DLA) process, and simulations of this were qualitatively in agreement with the TEM observations (see Supplementary Information Numerical simulation of the formation of nanoparticle aggregates)[77,78].

To understand if the improved flux associated with increasing NP ratio relative to enzyme concentration (Fig. 3f, g) correlated with increasing cluster size, differing amounts of 520 QDs and NPLs were added to the 7E system (Opt 2) at constant enzyme concentration yielding assemblies with low, medium, and high protein concentrations relative to NP. Supplementary Figs. 37, 38 show representative TEM micrographs of 520 QDs and NPLs assembled with these protein

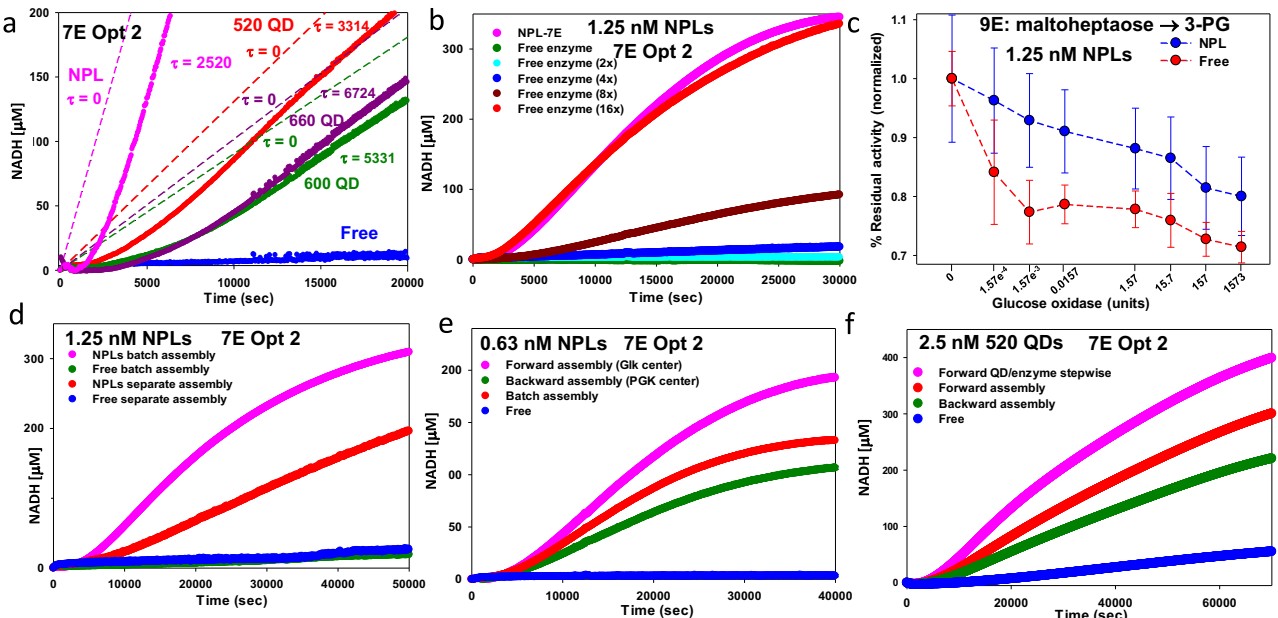

**Fig. 5 | Channeling phenomena. a** Estimated apparent transient times (τ, secs) from 7E cascade applied to QD/NPL data of Fig. 3c. Time-dependent product generation shown as solid plots. Linear region (except for the free enzyme, which had none) best fit with regression to determine τ (x-intercepts). 520 QD τ = 3314 ± 26, 600 QD τ = 5331 ± 64, 660 QD τ = 6724 ± 54, NPL τ = 2520 ± 18 sec. Dashed lines show slopes from fits with τ = 0 assumed as obtained under maximum channeling. **b** NPLs with 7E cascade (Opt 2 ratios). Corresponding free enzyme assayed with samples containing 2×, 4×, 8×, and 16× enzyme amounts held at Opt 2 ratios. **c** NPLs assembled with 9E system (maltoheptaose→3-PG with 7E at Opt 2 ratios) and corresponding free enzyme assayed with increasing glucose oxidase. NADH turnover normalized for each point and percentage residual activity of the cascade shown. 1.25 nM NPL used with ratios of enzyme/NPL in Supplementary Table 2. Data shown is the mean from n = 3 independent experimental samples ± standard deviation. **d** NPLs assembled with 7E at Opt 2 ratios in batch (enzymes mixed together first followed by NP addition) or separately where each enzyme added to 1/7th the NPL amount at the same concentration and ratio present in batch and then combined together prior to the assay start; designated by "separate assembly". **e** NPLs assembled with 7E-Opt 2 ratios. The order of enzyme addition to NPLs varied as indicated. Forward assembly added NP first, followed by Glk, then PGI, PFK, etc. Backward added NP first, followed by PGK, then GPD, etc. The batch added all enzymes at the same time, followed by NP, then mixed. **f** 520 QDs assembled with 7E-Opt 2 ratios. Forward assembly, backward assembly, and free prepared as above. Forward QD/enzyme stepwise added 1/7th QD followed by the first enzyme (Glk), then 1/7th QD followed by the second enzyme (PGI), and so on with 10 min between addition. Reaction NAD⁺ = 1.13 mM. NP/QD concentration indicated.

ratios, respectively (enzyme ratios Supplementary Table 2). These confirm that at high protein/NP ratios, the cluster size is small, while at low protein ratio, the converse is seen. To directly link increased flux to larger cluster size, identical concentrations of the 7E system (Opt 2) were assembled with increasing 520 QD concentrations and assayed. As seen in Fig. 4c, NADH production at 25,000 s increased *ca.* eightfold when increasing from 0.63 to 25 nM QD present (40× increase in QD concentration). Furthermore, significant increases in flux are seen only when the QD concentration reaches 2.5 nM. Aliquots from each of these same assay samples then underwent TEM analysis. Figure 4d, e shows TEM images from the 0.63, 2.5, and 25 nM QD samples along with the corresponding cluster analysis (full data Supplementary Fig. 39). Average cluster size increased 3× from ~3 to ~9 QDs from lowest to highest QD concentration, while flux simultaneously increased eightfold. In conjunction with the results of Fig. 3f, g, this confirms that larger cluster size is associated with better overall catalysis for QDs and for NPLs, presumably by incorporating more enzymes at higher density in that cluster sample. Why the effect saturates at high QD/NPL concentrations is less clear, but perhaps at high concentrations and large cluster sizes, the internal enzymes contribute less because their access to substrates/intermediaries is limited.

### Evidence for channeling in the NP-enzyme clusters

The presence of channeling in the NP-assembled cascades was confirmed using various classical experimental formats[11,29,72,79]. The first test estimated transient time (τ) by regression from the progress curves (solid lines) in Fig. 5a for the QD/NPL-7E system of Fig. 3c. The τ

of a sequential reaction is an observable lag in a reaction time course and is the time required for a cascade intermediate to reach a steady-state flux[29]. Transient times reflect channeling since in the strong channeling limit, they should go to zero, as depicted in Fig. 5a (dashed lines). In the opposite no-channeling limit, there will instead be a considerable transient delay as sufficient quantities of the various intermediates are being generated. For a MM cascade with N intermediates, τ is described by[80]:

$$\tau = \sum_{j=1}^{N} \tau_j \tag{2a}$$

where

$$\tau_j = \frac{K_{M,j}}{V_j - v_0} \tag{2b}$$

where $K_{M,j}$ and $V_j$ are the Michaelis constant and $V_{max}$ of the *j*th step (in which the *j*th intermediate is a reactant), respectively, and $v_0$ is the first enzyme's velocity. From Fig. 5a, the transient time estimate is lowest for the NPL system (~2520 sec), then increases in order of QD size (3314, 5331, and 6724 s for the 520, 600, and 660 QDs, respectively), qualitatively matching the improved rates for the different NPs (Fig. 3c and the average cluster sizes of Fig. 4a, b). The free-enzyme control did not have a clear linear regime so its transient time could not be characterized experimentally; simulation using Table 2 constants and assuming no-channeling estimated a τ of *ca.* 12,000 s.

Since channeling significantly enhances catalytic flux during the lag phase in a diffusion-limited regime, one should be able to match this behavior in the free-enzyme control by simply increasing enzyme concentrations to a sufficient point[27,29]. This is demonstrated in Fig. 5b, where NADH production for the NPL-7E system is compared to that of the corresponding free enzyme control. When enzyme concentrations are the same (Opt 2), the difference in production rate is many orders of magnitude; however, when concentrations in the free enzyme control are raised by 2×, 4×, 8×, and 16×, NADH production rates rise with the higher free enzyme concentration roughly matching the NPL configuration in turnover. Another classical confirmation of channeling presence is to catalytically challenge a cascade intermediate from the bulk environment[29]. A competing enzyme that consumes an intermediate is added to the solution with the expectation that this should perturb the cascade less when channeling is present as a result of the competitor's limited access to the channeled intermediate, especially if that molecule is ensconced in some type of enzyme aggregate structure[21,29]. To test this, we extended the 7E system to 9E by adding Amy and Mlt as upstream enzymes with maltoheptaose as substrate (Fig. 2). Turnover of the 9E free enzyme control was compared with the enzymes assembled on NPLs (ratios Supplementary Table 2). We used increasing concentrations of freely-diffusing GOx as a competitor targeting the glucose intermediate. Figure 5c plots normalized NADH production at 30,000 s vs. added GOx concentration present. Competing GOx has a much stronger inhibitory effect on the free enzyme control than the NPL-assembled system supporting the notion that channeling in the latter protects the glucose from being consumed.

A further test consisted of looking for reductions in turnover when proximity between the participating enzymes is disrupted. We compared results from two different assemblies of the 7E system on NPLs; in one, the enzymes and NPLs are assembled together in one "batch" aliquot as done above, while in the other, a "separate" assembly is used in which each enzyme is premixed with 1/7th of the total NPL and then combined together just prior to the assay start to minimize subsequent cluster formation. Figure 5d shows that separate assembly does indeed significantly reduce the 7E catalytic rate as compared to batch assembly into clusters. Still, another test looked for effects from vigorous shaking on the turnover with the expectation that shaking might reduce channeling by dispersing intermediates to the surrounding media at a much higher rate than in the static experiment. This approach was utilized previously as a test of channeling in the Pyk-LDH QD system[55]. When applied to the 7E system assembled on either NPLs or 520 QDs, shaking did indeed significantly reduce the catalytic rate in both (Supplementary Fig. 50).

Lastly, we sought to ascertain if enzyme ordering in the NP assemblies played any role since, as mentioned, there is debate over whether proximity or order is more influential in inducing channeling in such nanoscale structures[22,23,29]. We added the 7E system (Opt 2) to the NPLs in sequential order, with 10 min between each addition. "Forward" ordering matched the enzyme reaction sequence starting with Glk, then PGI, FPK, etc., while "backward" ordering started with PGK, then GPD, TPI, etc. Presumably, forward ordering had a higher probability of placing GLK at the cluster center with PGI proximal to it and then PFK further out, and so on, while backward ordering would place the cascade's terminal PGK centrally and put GLK at the cluster's periphery. Assays on these "ordered" NP-conjugates were compared with results from the NPL batch assembly used before and from free enzyme controls. Figure 5e reveals the forward configuration to have the highest overall catalytic flux, the backward to have the lowest, and the batch to have an intermediary value. Extending this, we undertook a modified "forward-stepwise assembly" where 1/7th of the total 520 QD was added, followed by Glk, then another 1/7th QD, then PGI, QD, FPK, etc. (Fig. 5f). This stepwise assembly provided for even more improvement in flux over that of just adding the enzymes in order

presumably by allowing for more enzyme incorporation into the nascent clusters. Other ordered assemblies were tested, including those based on enzyme size or ratio, $k_{cat}$, $K_M$, and $k_{cat}/K_M$; see Supplementary Fig. 50 for more on these results. Cumulatively, the various tests described here provide strong evidence in favor of substrate channeling being an important contributor to the enhanced catalytic rate observed in the nanoclustered assemblies. Unexpectedly, the ordering of enzyme addition and presumably concomitant ordered presence in the aggregates also appear to be a significant factor in determining overall turnover.

## Extended enzymatic cascades and coupling of channeled modules

Subsequent experiments focused on increasing the cascade size. We utilized the above 'forward' stepwise assembly process and added either Inv (8E, Fig. 6a) or Amy/Mlt (9E, Fig. 6b) upstream with sucrose or maltoheptaose as initial substrates, respectively. In all cases, enhancements in turnover were observed, with the NPLs performing best, especially in the 9E case. When Amy/Mlt were kept on separate QDs, the observed rate was slower than that seen when they were assembled along with the other seven enzymes in the same cluster. In Fig. 6c, Amy/Mlt were either assembled within the same QD cluster or separately assembled to their own 2/9th concentration cluster as QD concentration was varied to yield a 9E configuration. As two separate "modules", twice as much QD (i.e., 12.5 nM QD and Amy/Mlt QD) was required to increase the reaction rate (presumably by raising the average cluster size) to match that of the single clustered configuration (same 6.25 nM cluster). Pertinently, this result can also be considered another experimental proof of channeling presence. Inv and Amy/Mlt were sequentially combined in Fig. 6d to evaluate the contribution of each, and then all feeding glucose into the downstream 7E portion of the cascade. Interestingly, adding Inv to Amy/Malt in a 10E configuration slightly decreased the initial catalytic rate. Inv is the slowest enzyme and could be rate-limiting. However, glucokinases can also act on fructose[81]; thus, it is not clear if liberated fructose acted as a competitive inhibitor or if other factors caused this slight decrease.

The 4E system converting 3-PG to lactate was also assembled into nanoclusters demonstrating channeling behavior, which could again be augmented by increasing 520 QD concentration relative to enzyme (Fig. 6e). Attempts to extend the 7E system to create an 11E system processing glucose all the way to lactate by appending the 4E downstream steps did not show any channeling behavior despite producing some lactate as verified by mass-spectral analysis. Subsequent examination of each enzyme's $K_M$, predicted ΔG values, and concentrations of key intermediaries actually present in reactions (via mass-spectral analysis) suggested a complex interplay of underlying factors was responsible. As shown in Supplementary Fig. 12, and expanded on in its accompanying text, a sufficient initial concentration of intermediaries is produced in the upper portion of the first 7E system to overcome the high $K_M$ and positive ΔG of TPI and GPD. PGK presence should prevent GPD back-catalysis along with the benefits of the nanocluster providing enzyme proximity. We do not consider backward gluconeogenic reactions in this scenario since assaying 3-PG substrate in a PGK→GPD→TPI reverse cascade showed no activity (Supplementary Fig. 49). By the time the reactions reach the downstream 4E portion not enough 3-PG is being produced (ca. nM) to overcome the high $K_M$ (ca. 3.3 mM) and smaller predicted ΔG values of PGM and Eno to allow channeled flux to take place, even though the ΔG of the last two PykA→LDH steps are very favorable. We thus tested another strategy for merging the 7E-4E cascades as functional modules. In Fig. 6f, 3-PG produced from a first set of reactions (first module) by the 7E system using 520 QDs (Opt 2) was purified via HPLC (Supplementary Information), and then used as an initial high-concentration substrate bolus to initiate the 4E system assembled with 520 QDs in a second reaction (second module). This was quite effective, with both cascaded

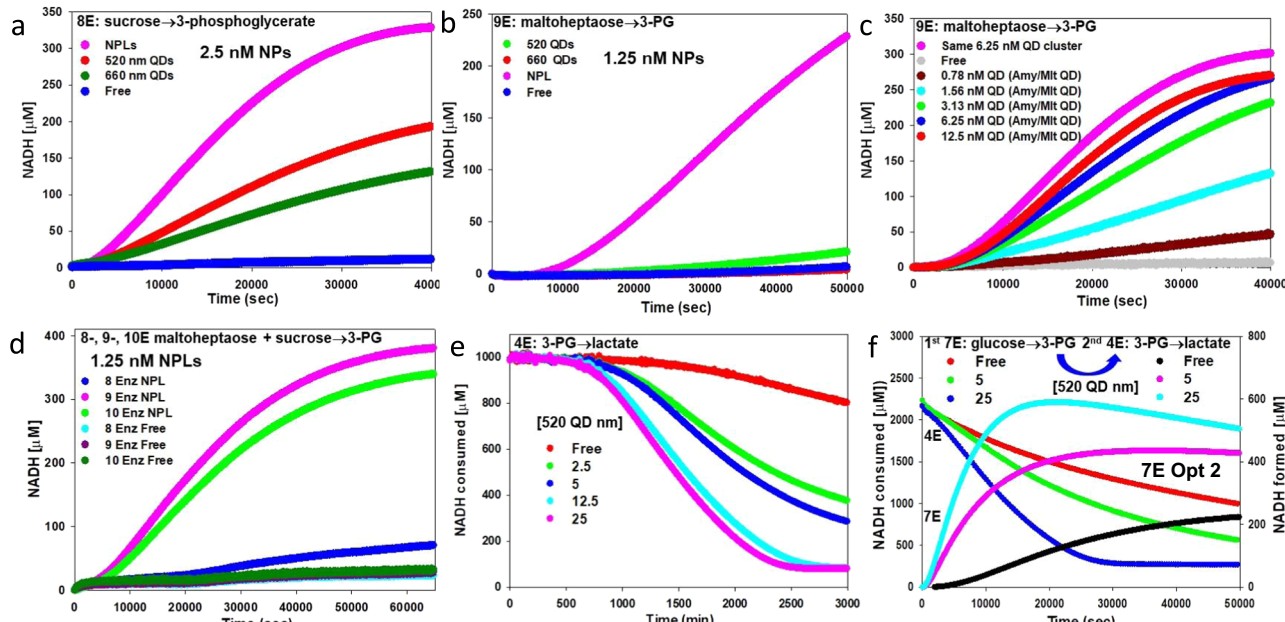

**Fig. 6 | Catalytic performance in functional configurations for 4, 8, 9, and 10 enzyme cascades. a** 520 QD, 660 QD, and NPLs with optimized 8E cascade converting sucrose→3-PG and for **b** optimized 9E cascade processing maltoheptaose→3-PG. **c** Varying 520 QD concentration and configuration with 9E cascade. Amy/Mlt added to 7E cascade to yield 9E. Enzymes (Amy 6.4, Malt 8.5, Glk 7.5, PGI 2, PFK 10, FBA 12, TPI 1, GPD 27, PGK 9.5 nM) assembled on the same QD cluster (6.25 nM pink), as a free enzyme (gray), or with increasing QD concentrations from 0.78 to 12.5 nM where Amy/Mlt each assembled separately to their own QD cluster at 1/7th total QD concentration and added to the 7E cluster. **d** 1.25 nM NPLs assembled with 8E cascade (7E + Inv), 9E cascade (7E + Amy/Mlt), 10E cascade (7E + Inv/Amy/Mlt), and free enzyme controls assayed with 120 mM sucrose and 4 mM maltoheptaose substrate. Ratios of enzyme/NPL in Supplementary Table 2. **e** Progress curves measuring NADH consumption for clusters with increasing 520 QDs added to fixed concentration 4E cascade converting 3-PG→lactate (PGM 18, Eno 8, PykA 19, LDH 19 nM). **f** Progress curves highlighting subsequent functional processing of 3-PG product from 7E cascade with 4E cascade converting 3-PG→lactate. 3-PG was initially produced in 96 well microtiter plates with fixed 7E-Opt 2 ratios mixed with 520 QDs at indicated concentrations for the 7E first reaction, monitored by NADH formation (right axis). 3-PG product was then purified using HPLC. Concentrated 3-PG produced from the first reaction was used as a substrate for the second reaction where the 4E cascade at a fixed concentration (PGM 18, Eno 8, PykA 19, LDH 19 nM) was assembled with 520 QDs at indicated concentrations in the second reaction. This reaction processed 3-PG to lactate as monitored by NADH consumption (left axis). NAD+ concentration 2.5 mM in panels **a**–**d**, **f**. NADH concentration in panels **e**, **f** 1.5 mM. In panel **f**, Some NADH was introduced as a carryover with co-purified 3-PG. Samples assembled using the forward process of Fig. 5e, f.

modules manifesting channeling and the rate of flux in both augmented by increased QD presence.

## Channeling in other self-assembled nanomaterial-enzyme clusters

To ascertain if the same self-assembled NP-enzyme clustered approach could extend access to channeling to other cross-linking nanomaterials beyond the QDs and NPLs utilized here, we undertook similar proof-of-concept experiments with other scaffolding materials, including AuNPs, commercially available QDs, and a commercially available dendrimer, see Fig. 7a, d, g. A key consideration was to maintain the same type of NP-enzyme assembly chemistry; hence all three of the tested materials display either NTA groups or multiple carboxyls that can bind Ni²⁺ and allow for the same type of metal-affinity coordination of the His₆ residues, albeit to a chelated metal as opposed to Zn²⁺ ions displayed as an intrinsic part of the QD shell.

Figure 7a shows representative TEM images of the *ca.* 5 nm diameter AuNPs utilized, which were surface-functionalized with 50% thioctic acid (TA)−50% TA-NTA (structures Supplementary Fig. 1)[52,82–84]. This NP diameter was chosen since it is close to the size of the better-performing 520 nm emitting QDs (4 nm diameter) and these AuNPs should display ≥50 NTA groups on their surface to coordinate the enzyme's His₆-motifs[65,85]. As an initial check, the AuNPs were first preloaded with Ni²⁺ and the assembly of the seven enzymes was confirmed by observing mobility shifts during separation in agarose gels, see Supplementary Information Enzyme assays and Supplementary Fig. 31. Results of enzyme assays are presented in Fig. 7b where the

activity of equivalent 5 nM concentrations of 520 QDs and 5 nM AuNPs each assembled with the 7E system at Opt 2 ratios are compared to free enzyme controls. The AuNPs are seen to elicit a similar enhancement effect on catalytic flux as the 520 QD assemblies but only at half the rate of the QDs. Exposing the 7E proteins to just the same concentration of Ni²⁺ alone (no AuNP control) appeared to be slightly inhibitory to the rate of flux. Figure 7c shows data from assays where the concentration of the 7E system was held constant while increasing amounts of the 5 nm AuNPs were added to form the assemblies, similar to that in Fig. 4c. The rate of flux increases from 2.5 to 5 nM AuNP concentration and then steadily decreased from there with concentrations of 25, 37.5, and 50 nM performing worse than the free enzyme. We note that commercial NTA-functionalized AuNPs in various sizes are available from several vendors, including Nanoprobes, Sigma-Aldrich, Nanopartz, and others.

We next examined commercial 525 nm emitting ITK carboxyl QDs. These QDs are surface-functionalized with a proprietary amphiphilic block copolymer that displays numerous carboxyl groups on the outer layer surrounding the QD[86]. Previous reports have confirmed that this carboxylated ligand can chelate Ni²⁺ and coordinate His₆-appended proteins in a manner somewhat akin to ZnS-overcoated QDs; these materials were similarly preloaded with Ni²⁺ ions and tested for self-assembly to the enzymes by agarose gels prior to the experiments (Supplementary Fig. 32)[62,87,88]. The hard diameter of these quasi-spherical QDs is *ca.* 5.7 ± 0.7 nm (Fig. 7d), similar to the AuNPs and 520 QDs, although the size of the surface copolymer is expected to more than double the hydrodynamic diameter of these materials as

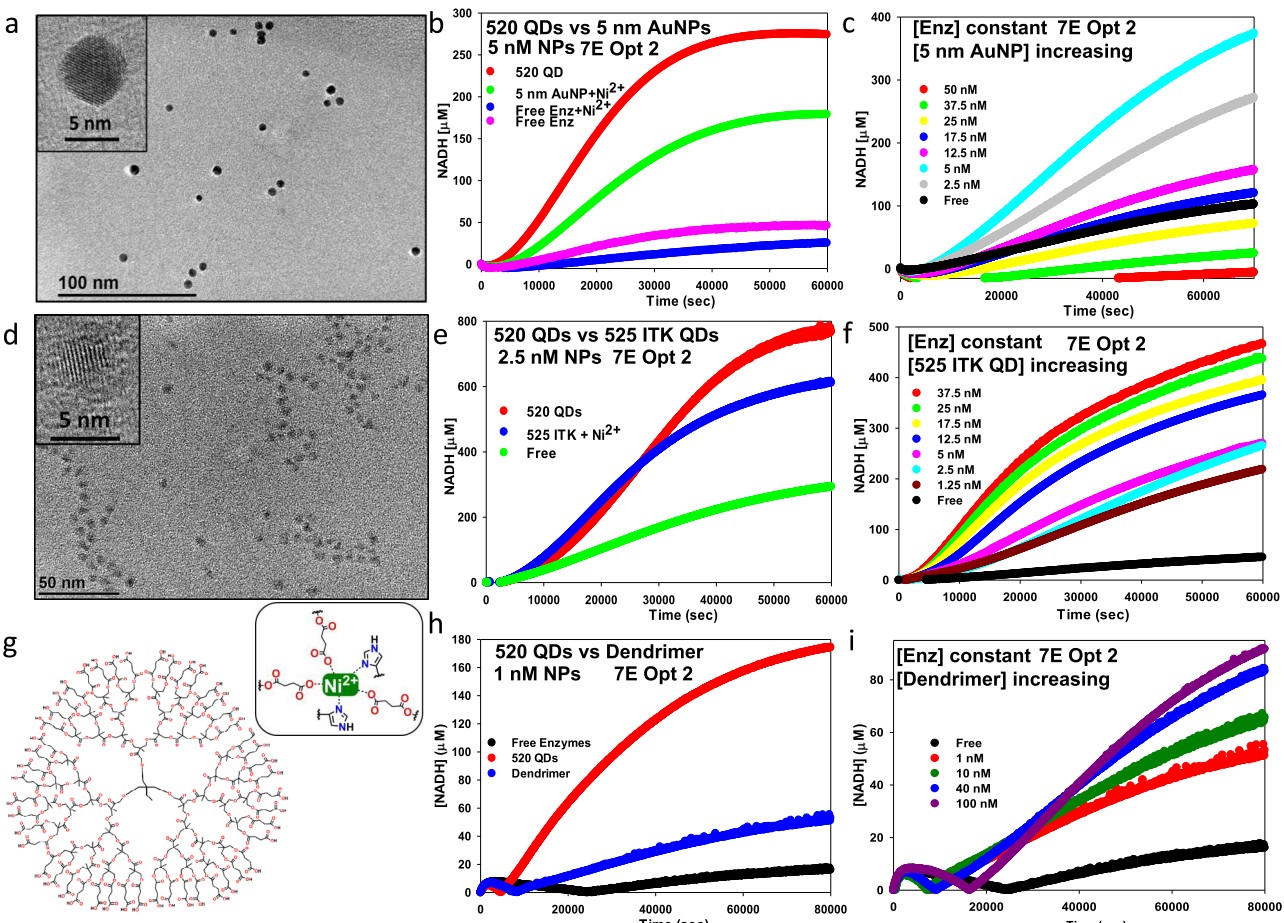

**Fig. 7 | Extending channeling in other nanomaterial-enzyme clusters. a** TEM micrograph of the 5 nm AuNPs (dia. 5.9 ± 0.7 nm). Inset shows a high-resolution micrograph of a single AuNP where a lattice structure is visible. **b** Representative progress curves comparing the activity of 5 nM 520 QDs and 5 nM Ni$^{2+}$-supplemented 5 nm diameter AuNPs preassembled with 7E system with the same concentration of enzyme versus free enzyme control. **c** Progress curves comparing 7E activity assembled at Opt 2 ratios versus the indicated increasing concentrations of Ni$^{2+}$-supplemented 5 nm diameter AuNPs preassembled with the same fixed concentration of enzymes. **d** TEM micrograph of the commercial 525 nm emitting ITK carboxy QDs (dia. 5.7 ± 0.7 nm). Inset shows a high-resolution micrograph of a single QD with a lattice structure visible. **e** Progress curves comparing the activity of 2.5 nM 520 QDs and 5 nM Ni$^{2+}$-supplemented 525 ITK carboxy QDs preassembled with 7E enzymes system at the same concentration of enzyme versus free enzyme control. **f** Progress curves comparing the 7E activity assembled at Opt 2 ratios versus indicated increasing concentrations of Ni$^{2+}$–525 ITK carboxy QDs preassembled with the same fixed concentration of enzymes. **g** Chemical structure of the bis-MPA-COOH dendrimer (trimethylol propane core, generation 4). Inset shows a close-up of 3 carboxyl groups chelating Ni$^{2+}$ in a manner analogous to NTA with the 2 imidazole side chain groups from histidine residues coordinating to the Ni$^{2+}$ by metal affinity. **h** Representative progress curves comparing the activity of 1 nM 520 QDs and 1 nM Ni$^{2+}$-supplemented dendrimer preassembled with the 7E system at the same fixed concentration of enzyme versus free enzyme control. **i** Representative progress curves comparing the 7E activity assembled at Opt 2 ratios versus the indicated increasing concentrations of Ni$^{2+}$-supplemented dendrimer preassembled with the same concentration of enzymes. Detailed experimental formats are in the Supplementary Information.

compared to the CL4-coated 520 QDs[70,86]. A comparison of relative fluxes for 2.5 nM 520 QDs and 525 ITK QDs assembled with the 7E system *versus* free enzyme is shown in Fig. 7e. Here, the initial rate of flux is equal between the two QD samples, although the 520 QDs produce a larger amount of NADH by the end of the assay timeframe. Figure 7f shows data from assaying constant concentration of the 7E system *versus* increasing concentrations of the 525 ITK QDs where increased flux correlates directly with the increase in QD concentration present. Lastly, to potentially extend beyond the use of hard NP materials to a softer scaffold, we also tested a commercial generation 4 carboxylated dendrimer with a trimethylol propane core (Fig. 7g). The performance of the previous two carboxylated NPs motivated this choice since the dendrimer displays 48 carboxyls at its outer periphery. As above, the dendrimer was preloaded with excess Ni$^{2+}$ ions and tested for enzyme assembly using PAGE analysis (Supplementary Fig. 33). Figure 7h compares the rate of NADH formation for 1 nM 520 QDs and 1 nM dendrimer assemblies versus free enzyme where the 7E dendrimer system performs at about 2× the rate of the free enzymes

but only a third that of the QD clusters. Figure 7i reveals that adding an increasing dendrimer to a constant 7E concentration also doubles the relative rate of NADH formation when going from 1 to 100 nM dendrimer.

For both the 5 nm AuNPs and 525 ITK QDs, TEM analysis confirmed nanoaggregate formation in the presence of the 7E system along with increasing aggregate size as a function of increasing the relative NP concentration (Supplementary Figs. 54, 55). This analysis also suggested that higher concentrations of AuNP in Fig. 7c led to super aggregate formation, which probably precipitated during the assays. For the AuNP and ITK QD, the data is quite analogous to that seen in Fig. 3 and Fig. 6, with our custom QDs/NPLs strongly suggesting a similar channeling process is at work. His$_6$ affinity for NTA (~1 µM) is not as strong as it is for the ZnS surface of QDs (~1 nM) and this difference could contribute to the somewhat poorer performance of the AuNPs and ITK QDs relative to the 520 ZnS-overcoated QDs[62,89]. For the dendrimer, the significantly reduced catalytic rates make the assignment of channeling to any catalytic improvements

more equivocal without significantly more work. Indeed, the high density of carboxyls in the dendrimer may create a highly-charged, localized environment where, similar to that seen with enzymes assembled to DNA scaffolds, substrate/intermediary sequestration can masquerade as channeling[24,44,45,48,50]. It is also probable that parametric testing of different carboxylated dendrimer analogs (e.g., with different generation number, different core design/scaffold, lower number of carboxyls, etc.) could identify one that performs far better.

## Discussion

Wheeldon's review on substrate channeling in enzymatic cascades provides an excellent overview of the current understanding in this field[29]. Channeling is primarily a nanoscale phenomenon and its advantages are obtained only when enzyme concentrations are in the diffusion-limited regime[72]. While inter-enzyme spacing is critical, the proximity of individual enzymes alone in a cascade is not sufficient for channeling outside cells[22,23,27–29]. When enzyme separation is greater than 1 nm, channeling requires some type of facilitation by bounded or confined diffusion. Wheeldon and others argue that solid evidence of channeling in most in vitro enzymatic systems studied to date is lacking, and what is needed are multiple methods of confirmation as well as evidence of the underlying structure that gives rise to the channeling mechanism[11,22,25,27,29]. It is even suggested that micro-environmental effects are often misinterpreted as channeling[45]. In work strikingly similar to ours, Mukai assembled the same Glk→LDH 10E cascade via His$_6$-tags to individual 500 nm diameter Ni-NTA silica NPs in different cascades using much higher enzyme concentrations that correspond to the bulk non-diffusion-limited regime[76]. In contrast to our results, they found no evidence of channeling with enzyme activity decreased in all cases when tethered to NPs. Large NP size may induce a localized stagnation boundary layer around the NPs that adversely affects diffusive processes and negates the enzymatic enhancements seen with smaller NPs[13,52].

One of the key precepts behind the assemblies described here, and that we argue is critical for both function and accessing channeling, is to minimize any chemical modifications or assembly strategies that deleteriously perturb native enzyme function. The highest catalytic rate of each participant enzyme is required and the ability to achieve this in a robust and predictable manner is also critical to accurately model the contribution of that enzyme within a (channeled) cascade using numerical simulations. For these reasons, we regard our exploitation of His$_6$ metal-affinity coordination as a key enabler. Importantly, such coordination relies on residues located at each enzyme's distal termini, which, in turn, allows all enzymes to assemble in the same non-perturbative manner, providing for increased enzyme activity in many cases and allowing us to undertake detailed modeling of each enzyme's channeled contribution to its given system's cascade in order to improve flux. In contrast, cross-linked enzyme aggregates (CLEAs) have been suggested as an alternative means of creating similar enzyme clusters without requiring an additional nanoscaffold[90,91]. Although extremely useful in many applications, it is well recognized that CLEAs commonly display reduced enzymatic activity due to the chemistry utilized in their assembly[90,91]. To ascertain if CLEAs could potentially function in a similar manner to our clusters, we selected tannic acid and Ni$^{2+}$ as two non-covalent assembly mechanisms along with glutaraldehyde as a third covalent chemistry for testing[92–102]. Initial assays (see Supplementary Figs. 51–53) confirmed that each of these agents did indeed deleteriously alter activity in the 7E system. Moreover, they modified the activity of individual enzymes in an unpredictable manner that varied between batch-to-batch preparations. Given the reductions in activity along with the issues of unpredictability (vis-á-vis performing numerical simulations), we did not pursue this system further. Reliance on His$_6$ metal-affinity coordination is not, however, limiting, and this was confirmed by our demonstrating that clustered assembly of enzymes and access to

channeling are achievable in three other NP systems, namely, AuNPs, dendrimers, and a commercially available QD with different surface chemistry.

Multiple layers of detailed evidence confirm that channeling is present in our NP-enzyme clusters (Fig. 5)[11,21,29,72,79]. All assays compared channeled formats to the corresponding free enzyme controls, and in all cases, the results are consistent with a channeling hypothesis. Though 7 of the 14-utilized enzymes manifest kinetic enhancement when displayed on NPs (Table 2), controls confirm that the enhanced-cascaded flux is not merely the sum of individual enhancements[51]. If this were true, batch vs. separate and ordered assembly in Fig. 5d, e would yield identical progress curves, as would separating the enzymes into two clusters (Fig. 6c). Previous modeling suggested that under efficient channeling conditions, the flux should increase sixfold for a two-step pathway and exceed 100-fold for a three-step pathway[11,25]. Depending upon the NP used, the number of enzymatic steps, and the reaction conditions/dilution, we observe maximum flux increases ranging from 60- to 100-fold. That these increases are not higher presumably stems from the random nature of the assembly processes, which produce NP-enzyme clusters where not every enzyme is necessarily present nor at an optimized ratio. Physico-chemical analyses of the cluster formation support this supposition (Supplementary Information). This non-ideality is exacerbated by the presence of high-turnover low-ratio enzymes such as PGI and TPI. We also found that assembly order and enzyme sequentiality can directly influence cascade flux contrary to prior modeling (though these models focused on much smaller cascades)[11,25]. Along with exploiting numerical simulations to optimize enzyme ratio, we showed that channeled flux can be further augmented through increasing cluster size by raising relative NP concentration. The latter serves to increase localized enzyme number/density as well as the probability of having all enzymes present in each cluster. This may explain the excellent performance of the NPLs, whose flat shapes (and intermediate S/V ratios relative to the QDs) presumably lead to more cross-linking and larger clusters. Based on kinetic and diffusional modeling, Wingreen predicted an optimal cluster size for one-step channeling to be ~260 nm[25], and interestingly our 7E NPL clusters assembled at low protein ratios approach this size (Supplementary Fig. 38). It appears that NP size/shape, enzyme ratio/assembly order, and cluster size can function as rudimentary control knobs over the rate of relative flux. Lastly, in contrast to reports of QDs interfaced with enzymes inside bacterial cells, we found no evidence for light-driven modulation of the kinetic flux when attempting to implement that format[103].

The channeling observed here is best described as probabilistic or proximity channeling and is clearly dependent on the clustered NP-enzyme structure. As we have shown, channeling occurs in these clusters because they are of sufficient size and enzyme density to form what Fernie terms an enzyme microdomain and Wingreen calls an enzyme agglomerate[11,25]. Within the agglomerate, the probability of an intermediate leaving an enzyme and encountering an individual downstream enzyme is low; however, the presence of so many colo-calized downstream enzymes greatly enhances the odds[25]. Critically, the substrate/intermediate concentration, $\Delta G$, and $K_M$ of the down-stream enzymes should all be favorable for the cascade to display channeling. As seen with attempts to append the 4E system at the end of the 7E system, channeling is no longer effective when downstream reactions become thermodynamically or kinetically unfavorable. In comparison to freely-diffusing enzymes in bulk solution, the NP-enzyme clusters might also function by providing soft-virtual com-partments that are defined by electrostatic and/or protein-protein interactions within the clusters[29]. Clusters may further prevent inhi-bitory substrates/products from interacting with some enzymes[11]. Though still largely uncharacterized, the NP surface, with its localized-structured microenvironment and nanoscale hydration layer, might also help facilitate channeling phenomena[104,105].

The self-assembled NP-enzyme clustering approach described herein represents a potentially powerful yet simple strategy for forming multienzyme cascades just through sample addition without the need for purification and without requiring complex molecular scaffolding[19,35,37,51,106,107]. The His$_6$ groups already present on the enzymes for purification are all that is needed for effecting self-assembly[108]. Furthermore, this approach overcomes issues associated with chemical cross-linking, which frequently inactivate enzymes. We recognize that Cd-containing QDs are not environmentally- or cost-effective scaffolds for larger scale applications and that other similar-sized materials such as ZnS NPs may be more suitable[109]. Our results support Wingreen's suggestion that, if properly configured, compact agglomerated enzyme clusters could ultimately provide similar advantages as directed channeling[25]. Channeled flux could possibly be increased even further by better control of cluster organization and internal enzyme stoichiometry; however, this seems a daunting challenge to implement given the number of variables involved, including NP size/shape, enzyme size/structure/shape, number of His$_6$-tags, etc. Preforming the NP clusters with designer cross-linking proteins prior to adding enzymes might offer a simpler alternative[110,111]. Accessing channeling in a similar manner can even benefit flux in larger, more-complex protein systems that are far less defined. For example, Thakur et al., demonstrated that the addition of similar CL4-functionalized QDs to a cell-free protein synthesis system where the constituent proteins all displayed His$_6$ residues at their termini could improve target protein production by up to 12-fold[112]. Along with providing a platform to help elucidate the benefits of channeling, these structures suggest themselves as possible components for artificial cells as they recapitulate some of the metabolic functionality of cells without requiring surrounding membranes[9,10,13]. They can also act as artificial metabolons for use in enzyme-based biosynthesis of non-natural products[113–117]. The ability to self-assemble enzyme clusters for small-volume, high-efficiency reactions and then screen them against many different substrates would certainly be beneficial in this regard.

## Methods
### Materials
**Reagents.** All chemicals, buffers, and reagents were sourced and utilized as supplied by standard vendors including Millipore, Sigma-Aldrich, and Thermo Fisher, unless specified. Where a specific non-standard item is used or was obtained from another vendor, that information is provided.

**Quantum dots.** The words QDs and NPLs are used to refer to the individual materials, while NPs is used to refer to them collectively. 520-, 600-, and 660 nm emitting CdSe/CdS/ZnS core/shell/shell QDs with average diameters of ~4.0 ± 0.4, 9.7 ± 1.0, and 13.4 ± 1.3 nm, respectively, were synthesized as described in detail in refs. [68,118]. QDs were cap-exchanged with the zwitterionic dihydrolipoic acid- (DHLA) based Compact Ligand CL4, the chemical structure is shown in Supplementary Fig. 1[70]. This ligand provides for long-term QD colloidal stability in buffer and challenging environments such as cells and tissues while still allowing polyhistidine metal-affinity coordination of enzymes to the QDs surface[62,63,119–121]. QD size was confirmed with transmission electron microscopy (TEM) analysis as described in ref. [122].

### Nanoplatelets
**Synthesis of 4 monolayer (4ML) CdSe nanoplatelets.** CdSe NPLs were synthesized according to published methods with slight modification[123]. Cadmium myristate · Cd(myr)$_2$ (170 mg, 0.3 mmol), Se (12 mg, 0.15 mmol), and octadecene (ODE) (15 mL) were combined in a 50 mL three-neck round-bottom flask. The mixture was degassed under vacuum for 30–60 min at 50 °C. Next, under nitrogen, the heating mantle was set to 240 °C. When the solution turned orange, around 190–200 °C, powdered cadmium acetate−Cd(Ac)$_2$·2H$_2$O (80 mg, 0.3 mmol) was added. After 9 min at 240 °C, the heating mantle was removed, and the reaction was quickly cooled to room temperature. During the cooling process, 1 mL of degassed oleic acid was injected at 100 °C. At room temperature, 12 mL of hexane and 12 mL of ethanol was added, and the platelets were precipitated by centrifugation. The supernatant was discarded and the orange pellet dissolved in 10 mL of hexane. The hexane solution is centrifuged and the supernatant containing the 4 ML CdSe NPLs separated from the yellow pellet. The NPLs are stored in hexane at room temperature in the dark. NPL structure and size was confirmed by TEM[122].

**Synthesis of CdSe/ZnS core/shell NPLs.** ZnS shell growth on the 4 ML CdSe NPLs was achieved with procedures published by ref. [124]. CdSe NPLs (0.452 nmol) in hexane was diluted with 2 mL of toluene in a 20 mL scintillation vial or small round-bottom flask. The hexane was gently removed under reduced pressure. Next, 35 μL of 0.5 M ZnCl$_2$ in oleylamine (preheated to 50 °C), 35 μL of trioctylphosphine (TOP), 3.5 μL CS$_2$, and 6.4 mg of Zn diethyldithiocarbamate−Zn(DDTC)$_2$ were added to the NPLs. The reaction mixture was heated to 110 °C for 1–2 h with gentle stirring and then cooled to room temperature. A minimal amount of isopropanol was added to destabilize the NPLs and they were precipitated by centrifugation. The supernatant was discarded and the core/shell NPLs were dissolved in toluene and stored at room temperature in the dark.

**Cap exchange of CdSe/ZnS NPLs with CL4.** The disulfide, methyl ester form of CL4 (126 mg, 0.3 mmol) was dissolved in ethanol (1 mL) and DI water (0.5 mL) and stirred with LiOH (16 mg, 0.67 mmol)[70]. After 1 h, the solution was adjusted to pH 7–8 by slowly adding 4 M HCl dropwise. Next, NaBH$_4$ (25 mg, 0.66 mmol) was added and the mixture was stirred for 1 h. After the solution turned colorless, 4 M HCl was added dropwise to adjust the pH to 7–8. Separately, a portion of CdSe/ZnS NPLs (2 nmol) in toluene were precipitated with minimal iso-propanol and centrifuged. The supernatant was discarded and the CdSe/ZnS NPLs were dissolved in 1 mL of chloroform and added to the activated ligand mixture with vigorous stirring. Small portions of chloroform and DI water were added until a biphasic mixture was achieved. The mixture was rapidly stirred until the NPLs were transferred to the aqueous phase (2–24 h). The organic phase was discarded and the aqueous phase was washed with CHCl$_3$ (3 × 1 mL). The aqueous phase was filtered through a 0.45 μm hydrophilic membrane filter (Millipore) and washed with DI water (2–3 × 1.5 mL) using a centrifugal filtration device (Millipore, MW cutoff 100 kDa). The aqueous CL4-capped NPLs were stored at 4 °C in the dark until further use. The final NPL material had an emission at ~585 nm (585 NPL).

**Gold nanoparticles.** About 5 nm AuNPs surface-functionalized with 50% TA/50% TA-NTA were synthesized as described in detail[52,82,85,122,125,126].

**Qdot 525 innovators tool kit (ITK) carboxyl quantum dots.** About 8 μM solution catalog number: Q21341MP were obtained from Invitrogen by Thermo Fisher Scientific.

**COOH dendrimer.** bis-MPA-COOH dendrimer trimethylol propane core, generation 4 catalog number: 806072, was obtained from Sigma-Aldrich.

### Enzymes
**Gene sequences and cloning.** Genes for α-amylase from *B. cereus*, alpha-glucosidase from *Saccharomyces pombe* which we refer to as maltase[43,51], and INVB an invertase from *Zymomonas mobilis*[127], were codon optimized for expression in *E. coli* and synthesized by Genscript (protein sequences are listed in the Supplementary Materials). Genes

were subsequently cloned into the NcoI and XhoI restriction sites in expression vector pET28a (Addgene). DNA constructs were verified by sequencing performed by Eurofins Genomics and the plasmids were transformed into the *E. coli* BL21(DE3) strain for protein expression. The remaining enzymes were cloned directly from *E. coli* using custom primers that matched the N- and C-termini of their known gene sequences using PCR[55].

**Generalized protein expression and purification protocol.** Assembled plasmids were sequence confirmed following transformation and antibiotic selection in *E. coli* strain DH5a (New England Biolabs, Ipswich, MA, USA). Plasmid DNA from sequence-confirmed clones was transformed to *E. coli* strain BL21(DE3) (New England Biolabs, Ipswich, MA, USA) for bacterial expression. Single colonies from antibiotic selection plates (LB plus 50 µg/mL kanamycin) were inoculated to liquid broth, grown to mid-log stage, then combined with sterile glycerol to a final concentration of 40% v/v to prepare glycerol stocks, which were stored at −80 °C.

Protein expression and purification utilized the following general procedure and have been described in detail in other references[52,55,56]. Glycerol stocks were streaked to isolate single colonies on LB plus kanamycin plates. Overnight cultures were started from single colonies and grown overnight at 37 °C. The following morning, 500 mL of Terrific Broth (TB, Sigma-Aldrich, USA) containing kanamycin in a 2 L baffled flask was inoculated with 5 mL of the overnight culture. Flasks were incubated at 37 °C and shaken at 250 rpm for 3 h or until mid-log (OD$_{600}$ = 0.6–0.8) was attained. Production was initiated through the addition of 0.5 mM isopropyl-β-D-thiogalactopyranoside (IPTG) and maintained for 12–16 h. The temperature was lowered to 30 °C post-IPTG addition while shaking was maintained at 250 rpm. Cell suspensions were transferred to polypropylene, screw-top bottles, and centrifuged for 15 min at 4000×$g$ to pellet cells. Pellets were weighed and then transferred to −80 °C freezers to await further processing (minimum storage of 12 h at this temperature).

Cell pellets were thawed on ice then resuspended in lysis buffer (1/2x phosphate buffered saline, 1 mM EDTA, 1 mg/mL hen egg white lysozyme, 0.1% Triton X-100) and incubated on ice for 30 min with periodic mixing through inversion. Following incubation on ice, samples were sonicated using a Branson Sonifier at 90% amplitude, cycle 0.5, and 60 s intervals. A minimum of three cycles were used to ensure cell lysis. Lysates were transferred to Nalgene 50 mL Oak Ridge style 3119 tubes (Sigma-Aldrich, USA) and centrifuged at 4 °C and 12,000×$g$ for 20 min to pellet cell debris. Soluble material was decanted to a 50 mL Falcon tube (Fisher Scientific, USA) and placed on ice. A 750 µL aliquot of immobilized metal-affinity chromatography (IMAC) resin (Ni Sepharose High Performance, Sigma-Aldrich, USA) was transferred to a microfuge tube then equilibrated in column wash buffer (50 mM phosphate pH 6.0, 300 mM NaCl, 25 mM imidazole, 0.05% Tween-20) using a batch wash method. Equilibrated resin was added to the soluble protein fraction and the entire suspension was equilibrated through addition of the stock wash buffer solution (prepared at 5x strength) which was diluted to a final 1× concentration. The Falcon tubes were transferred to a rotary wheel and incubated for a minimum of 3 h at 4 °C. Resin was batch washed in the Falcon tubes using low speed centrifugation (400×$g$) and cold column wash buffer. Resin was washed with a minimum of 60-bed volumes using this method then transferred to a gravity chromatography column (9 cm Poly-Prep Chromatography Columns, Bio Rad, USA). Captured proteins were eluted with wash buffer containing 300 mM imidazole. Fractions were collected in 1 mL aliquots, which were stored on ice. Protein-containing fractions were identified via measurement of absorbance at 280 nm using a Nanodrop One Microvolume UV-Vis Spectrophotometer (Thermo Fisher Scientific, USA) and examined for purity via SDS-PAGE on 4–15% gradient Tris-glycine gels (Bio Rad, USA). Enzyme-containing fractions were pooled and dialyzed against 50 mM phosphate buffer

(pH 8.0). Enzyme concentration was determined by UV-Vis measurement of their absorbance using their predicted extinction coefficient. Enzymes samples were supplemented with 25% glycerol prior to aliquoting to 0.5 mL microfuge tubes for snap freezing in a dry ice-methanol bath and final storage at −80 °C. For assays, individual tubes were removed from storage, thawed, used, and any remaining enzyme discarded.

## Assays

**Enzymatic assays—overall approach.** The apparent kinetic parameters for each of the 14 enzymes associated with glycolysis and saccharrification were determined individually. A detailed individual methodology for 13 of the 14 enzymes is described in the Supplementary Information with the methodology for pyruvate kinase A found in ref. [55]. A more general overall method for determining kinetic parameters is described here. Three different types of assays were used to determine the apparent kinetic parameters: measuring the absorbance of NADH formation or its consumption at 340 nm (ε = 6220 M$^{-1}$ cm$^{-1}$), utilizing a commercially obtained assay kit, or a coupled assay format where excess downstream enzyme(s) was used and coupled NADH formation/disappearance was measured[21]. For the coupled assay formats, all enzymes and co-substrates were added to the reaction in excess to be saturating. The amounts of each to use in an assay to reach a saturating condition were predetermined empirically with parameterized testing. Due to assay functional constraints, PGI and PGK activity were determined by using the opposite direction of activity relative to the overall catalytic flux. This approach is justified, and especially for isomerases, when such constraints are present as discussed in refs. [21,128].

Typically, a set concentration of enzyme was assembled to increasing molar ratios of QDs at double the final concentration in buffer unless otherwise noted. These assays utilized a range of QD concentrations to give typical enzyme-to-QD ratios of 0, 0.25, 0.5, 1, 2, 4, 8, and 12. The results from these assays (see listings Supplementary Tables 14–34) were used to determine if enzymes displayed kinetic enhancement when assembled with QDs. If enzymes did show enhancement with spherical 520 nm emitting QDs (diameter ~4 ± 0.4 nm), then they were also typically tested with NPLs across a similar range of enzyme-to-NPL ratios. Prior to actual assay reactions, the assemblies were allowed to incubate at 4 °C to ensure appropriate enzyme-QD bioconjugate formation for at least 30 min. The enzyme-QD bioconjugates were dispensed to a 384-well plate which was then briefly spun in a centrifuge to ensure no droplets clung to the side, similar in manner as described in ref. [63]. Before the start of the assay, substrate over a range of concentrations with or without excess coupled enzyme(s) and all appropriate salts and cofactors were dissolved in buffer. The highest substrate concentration had a final concentration that was usually at least four times or more the literature $K_M$ value as a starting point. The substrate solutions were added to the plate as quickly as possible in a similar manner as described in ref. [63]. The components of the substrate stock solutions were usually twice as concentrated as the final reaction concentration unless otherwise noted. Each enzyme-QD-substrate combination was performed in triplicate or quadruplicate. The plate was covered with a piece of film to prevent evaporation and placed in a Tecan Spark plate reader where a kinetic program was immediately started. In general, the kinetic program consisted of maintaining the temperature at 30 °C, shaking the plate for a few seconds to ensure adequate mixing before measuring the absorbance, and then continuously measuring the absorbance at a defined wavelength and time interval for at least 16 h unless otherwise noted. The progress curves were plotted in Excel and the slope of the linear portions of curves was determined for each initial rate. The substrate concentrations and corresponding initial rates were fitted to the Michaelis–Menten equation utilizing Sigma Plot®'s enzyme macro or by minimizing the error between the estimated initial rate and

actual rate when changing the $K_M$ and $V_{max}$ values. $k_{cat}$, enzyme specific activity (SA), and $k_{cat}/K_M$ were estimated using standard methods as described in refs. [21,51,55,75,129]. We stipulate that all values are apparent as it is not clear that the function of enzymes as attached to NPs or in NP clusters meet all basic Michaelis-Menten assumptions[54,74]. Nevertheless, these values are still reported here and specifically utilized as they are excellent comparators of enzyme activity when free in solution and when displayed on NPs.

## Multienzyme assays

**Upstream 7 (7E)–10 enzymes.** In general, the ratios of the enzymes to NP were varied as indicated in figure legends and tables. The order of assembly was also varied with the Forward reaction being NP added first followed by Glk then PGI, and so on until PGK for the 7E system. The Backward reaction was done by adding NP first followed by PGK, then GPD, and so on. Batch was done by adding Glk first followed by PGI, etc. and then finally adding NP after the solution had been lightly vortexed. In more detail, aliquots of the seven enzymes were thawed on ice and lined up in order in a centrifuge rack. Stock solutions were made by diluting aliquots of each enzyme in buffer (250 mM HEPES pH 8). The stock solution of NP was maintained at 10 nM and the concentration of each enzyme was adjusted by its ratio relative to NP concentration. For enzymes not assembled to a NP, an equivalent amount of enzyme that would have been mixed with the 10 nM NP was added to the control solution. After the final addition of each enzyme the solutions were wrapped in foil and placed at 4 °C for at least 3 h to allow for assembly to occur. Following assembly, buffer was added to bring the volume up to maintain a NP concentration of 10 nM. The solutions were diluted in half sequentially with HEPES buffer to achieve NP concentrations ranging from 10 nM down to 1.25 nM and the appropriate concentration of enzyme followed suit. 25 μL of each sample was added to a 384-well plate with each condition represented in triplicate or quadruplicate. The 384-well plate was briefly centrifuged at low speed for less than 30 s to ensure the sample volume was at the bottom of the well.

Prior to the start of an experiment, a fresh stock substrate solution was made each time. This consisted, for example, of 30 mM MgCl₂, 15 mM ATP, 15 mM ADP, 20 mM glucose, 8 mM dibasic/monobasic phosphate, 2.25 mM NAD⁺, and 250 mM HEPES. The phosphate solution was added last to ensure it does not react with the MgCl₂ to form a precipitate. To the 384-well plate, 25 μL was added to each well to start the reaction either with a 12-well pipette or the liquid dispensing device on the plate reader. The plate was immediately covered with protective clear film and placed in a Tecan Spark Microplate reader where the formation of NADH or its consumption was monitored over 18 h by measuring the absorbance at 340 nm every minute. The plate was briefly shaken for 3 s prior to each measurement to ensure homogenous mixing and the temperature was maintained at 30 °C unless otherwise stated. The final concentration of NP ranged from 0 to 5 nM with enzymes as a multiple of that concentration at the indicated ratios used while the final concentration of the rest of the components was 15 mM MgCl₂, 7.5 mM ATP, 7.5 mM ADP, 10 mM glucose, 4 mM dibasic/monobasic phosphate, 1.13 mM NAD⁺, and 250 mM HEPES. Absorbance measurements were converted to NADH concentrations formed or consumed utilizing the Beer-Lambert equation and an extinction coefficient of 6220 M⁻¹ cm⁻¹. A similar process was utilized for the 8, 9, and 10 enzyme systems when amylase, maltase, and invertase were added to extend the cascade while adjusting the ratios of enzyme per NP to that listed in Supplementary Table 2. Initial substrate concentrations of maltoheptaose and sucrose were 4 and 120 mM when assayed individually. When invertase, amylase, and maltase were combined to extend the 7 enzyme system to 10 enzymes, the concentration of maltoheptaose and sucrose was again 4 and 120 mM, respectively. Lastly, for testing of channeling phenomena, assays

were implemented in the same way with the indicated changes to format. We did not implement any isotope dilution or enrichment assays to test channeling as our institution currently does not allow working with radioactive biomaterials[11,29].

**Downstream 4 (4E) enzymes.** The last four enzymes of the glycolytic pathway studied here, namely phosphoglycerate mutase, enolase, pyruvate kinase, and lactate dehydrogenase, were assembled onto NPs separately from the 7E systems. The 4 enzyme (4E) construct was assembled in the forward direction, i.e., NP first, followed by PGM, Eno, PykA, and finally LDH at the indicated ratios. As with the 7E system, stock aliquots of enzyme were thawed on ice and lined up in order in a centrifuge rack. The following stock enzyme solutions were made in 250 mM HEPES buffer pH = 8, 7.50 μM PGM, 3.86 μM Eno, 9.36 μM PykA, and 9.40 μM LDH. A stock solution of 520 QDs at 0.65 μM was also made in the HEPES buffer. The final concentration of each enzyme was determined for the case when the final QD concentration was maintained at 5 nM. The concentration of each enzyme was adjusted by its ratio relative to this QD concentration. The concentration of each enzyme was maintained when there was no QD present or when the final QD concentration was 25 nM relative to the 5 nM case. After the final enzyme was added, the solutions were lightly vortexed, gently centrifuged, then wrapped in foil and allowed to assemble at 4 °C for at least 2 h. Afterwards, HEPES buffer was added to bring up the volume to maintain a QD concentration at 0, 10, or 50 nM. A stock substrate solution was made composed of 8 mM mono/dibasic phosphate, 30 mM MgCl₂, 20 mM ADP, 3 mM NADH, 15 mM 3-phosphoglycerate, and 250 mM HEPES buffer. To a 384-well plate, 25 μL of the enzyme bioconjugate was added followed by 25 μL of the substrate solution. The plate was briefly centrifuged for 30 s to ensure all droplets were at the bottom of the well. The consumption of NADH was utilized to follow the progress of this 4E cascade by measuring the absorbance at 340 nm over time. Given the much longer reaction time needed for this 4E system to consume the majority of NADH, the experiment was run for at least 4 days (>90 h) as opposed to the 16 h as with the 7 E system. Experiments were run on a Tecan Spark microplate reader, maintaining the temperature at 30 °C, shaking the plate for 5 s before reading the absorbance, and an interval time ranging from 5 to 6 min. The final concentration of all components in the well is as follows: 89.5 nM PGM, 38.5 nM Eno, 93.5 nM PykA, 94 nM LDH, 0 to 25 nM QD, 4 mM monobasic/dibasic phosphate, 15 mM MgCl₂, 10 mM ADP, 1.5 mM NADH, 7.5 mM 3-phosphoglycerate in 250 mM HEPES, pH 8.

## Reporting summary

Further information on research design is available in the Nature Portfolio Reporting Summary linked to this article.

# Data availability

The data that support the findings of this study are available within the main text and its Supplementary Information file. Source data is provided as Source Data file. Data were also available from the corresponding author upon request. Source data are provided with this paper.

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

## Acknowledgements

The authors acknowledge the Office of Naval Research (Award N0001421WX01580), the US Naval Research Laboratory (NRL), and the NRL Nanoscience Institute for funding support. J.N.V., S.L.H., and W.P.K. acknowledge NRC Fellowships through NRL. I.L.M. and G.A.E. acknowledge the National Institute of Food and Agriculture, U.S. Department of Agriculture, under Award #2020-67021-31254, and the Strategic Environmental Research and Development Program (SERDP), under Award # WP21-1073 New Start Project (W74RDV03497375).

## Author contributions

J.C.B., J.N.V., G.A.E., M.G.A., and I.L.M. provided initial concepts along with study/experimental design and project management. J.N.V., E.O., M.H.S., K.S., and S.A.W. synthesized and provided key materials and enzymes. J.C.B., J.N.V., G.L.A., G.A.E., S.A.D., S.L.H., W.P.K., M.T., and M.G.A. undertook experimental analysis and interpretation. J.C.B., J.N.V., E.O., S.A.D., M.T., and I.L.M. undertook assays and materials analysis. S.A.D., G.A.E., W.P.K., and M.G.A. wrote software and provided numerical simulations and other computerized analysis. All authors contributed data and writing and approved of the final manuscript.

## Competing interests

I.L.M., J.N.V., M.G.A., K.S., and S.A.D. received Patent No.: US 11,512,305 B2 entitled Nanoparticle-Attached Enzyme Cascades for Accelerated Multistep Biocatalysis, which was filed by the US Navy. This patent includes some of the processes and phenomena described herein. The remaining authors declare no competing interests.
