## [Peer Review File · Nature Communications]

Self Assembled Nanoparticle Enzyme Clusters Provide Access to Substrate Channeling in Multienzymatic CascadesREVIEWER COMMENTS

Reviewer #1 (Remarks to the Author):

Authors do not try to introduce immobilization. Modern enzyme immobilization, far to be only a way to ensure enzyme recovery and reuse, is nowadays a tool to improve many enzyme features, such as stability, activity, selectivity, specificity, inhibitions, even enzyme purification may be coupled. There are reviews in each of these topics, and at least this need to be mentioned. Because one of the problems of coimmobilization is that all enzyme are immobilized following the same strategy, that may be not optimal for all enzymes.

Moreover, as reviewed by Arana, enzyme coimmobilization has many other problems. They cannot ignore this in introduction, advantages are not the only point to be pointed. Not in all cases the use of individually immobilized/optimized enzymes can be discarded in favor of coimmobilization.

Authors make emphasis on enzyme proximity, when there are many reports that conclude that the main effect is the increment of intermedium concentration due to space limitations;

Zhang, Y., Tsitkov, S., Hess, H.

7601307821;57190762872;7201886138;

Proximity does not contribute to activity enhancement in the glucose oxidase-horseradish peroxidase cascade

(2016) Nature Communications, 7, art. no. 13982, . Cited 182 times.

[https://www.scopus.com/inward/record.uri?eid=2-s2.0-](https://www.scopus.com/inward/record.uri?eid=2-s2.0-85006982391&doi=10.1038/ncomms13982&partnerID=40&md5=46667cd79cf6c2c27f9735663564d1fb)

[85006982391&doi=10.1038/ncomms13982&partnerID=40&md5=46667cd79cf6c2c27f9735663564d1fb](https://www.scopus.com/inward/record.uri?eid=2-s2.0-85006982391&doi=10.1038/ncomms13982&partnerID=40&md5=46667cd79cf6c2c27f9735663564d1fb)

DOI: 10.1038/ncomms13982

Xiong, Y., Tsitkov, S., Hess, H., Gang, O., Zhang, Y.

57208566514;57190762872;7201886138;6701826437;7601307821;

Microscale Colocalization of Cascade Enzymes Yields Activity Enhancement

ACS Nano, . In press DOI: 10.1021/acsnano.2c00475

Idan, O., Hess, H.

37124124000;7201886138;

Origins of activity enhancement in enzyme cascades on scaffolds

(2013) ACS Nano, 7 (10), pp. 8658-8665. Cited 92 times.

[https://www.scopus.com/inward/record.uri?eid=2-s2.0-](https://www.scopus.com/inward/record.uri?eid=2-s2.0-84886995368&doi=10.1021/nn402823k&partnerID=40&md5=db9d6dc8e376931f6222d42f7cf6749b)

[84886995368&doi=10.1021/nn402823k&partnerID=40&md5=db9d6dc8e376931f6222d42f7cf6749b](https://www.scopus.com/inward/record.uri?eid=2-s2.0-84886995368&doi=10.1021/nn402823k&partnerID=40&md5=db9d6dc8e376931f6222d42f7cf6749b)

DOI: 10.1021/nn402823k

DOCUMENT TYPE: Article

PUBLICATION STAGE: Final

SOURCE: Scopus

Idan, O., Hess, H.

37124124000;7201886138;

Engineering enzymatic cascades on nanoscale scaffolds

(2013) Current Opinion in Biotechnology, 24 (4), pp. 606-611. Cited 57 times.

[https://www.scopus.com/inward/record.uri?eid=2-s2.0-](https://www.scopus.com/inward/record.uri?eid=2-s2.0-84880960930&doi=10.1016/j.copbio.2013.01.003&partnerID=40&md5=99dd41dcd19ad3e1d096fc8a8bfe2664)

[84880960930&doi=10.1016/j.copbio.2013.01.003&partnerID=40&md5=99dd41dcd19ad3e1d096fc8a8bfe2664](https://www.scopus.com/inward/record.uri?eid=2-s2.0-84880960930&doi=10.1016/j.copbio.2013.01.003&partnerID=40&md5=99dd41dcd19ad3e1d096fc8a8bfe2664)

DOI: 10.1016/j.copbio.2013.01.003

DOCUMENT TYPE: Review

PUBLICATION STAGE: Final

SOURCE: Scopus

Idan, O., Hess, H.

37124124000;7201886138;

Diffusive transport phenomena in artificial enzyme cascades on scaffolds
(2012) Nature Nanotechnology, 7 (12), pp. 769-770. Cited 19 times.
<https://www.scopus.com/inward/record.uri?eid=2-s2.0-84874588554&doi=10.1038/nnano.2012.222&partnerID=40&md5=4aff3cb53ba41a916be9d01b6bbf1f3a>

DOI: 10.1038/nnano.2012.222

DOCUMENT TYPE: Letter

PUBLICATION STAGE: Final

They cited some papers where the authors show that the positive effects are due to other effects different to substrate channeling. However, in these reports, if the enzyme proximity is so important, this should be also a main factor in that papers.

If they want to make a deeper study on this, the paper could be a very good one,

To ensure that the substrate channeling is the key, they should combine sequentially immobilized enzymes (in form of crowns, where only the enzyme in the border of the crown are bear to the other sequential enzymes) using "logic and illogic" enzyme sequences, and enzyme colocalization, to confirm that the effect of the substrate channeling and not the most evident increase of substrate concentration for each enzyme.

In any case, the current paper seems to be a new step on a previously described strategy, including more enzymes. In my opinion, this not justify the publication in a highly reputed journal such as Nature communications.

Reviewer #2 (Remarks to the Author):

This study is quite in depth and highlights a relatively universal method for local enhancement of enzymatic cascades with commonly-used His-tag additions to proteins and readily available assembly materials. As the authors have stated, this work is meant to explore and clarify concepts that have been both theoretically and experimentally explored using different systems. Moreover, many systems have used particles/surfaces for assembly as well. The presented experimental design elucidates that as opposed to directly scaffolded systems with particular attention paid to spacing and enzyme order, there is significant, impactful benefit to more broadly-considered enzyme ordering and packing. The authors have expanded the enzyme systems explored in relation to other works by considering a larger and more complex cascade. Overall, this is an interesting, timely and important work which will be of interest to a broad community, thus, in my opinion, it might be suitable for Nature Communications. Several comments should be addressed, as listed below

The writing of the paper is quite dense, and figures beyond Figure 1 would benefit from at least minimal schematics, as each is tied together conceptually but doesn't have illustrative aspects defining the core of this paper, which is assembly considerations. The aggregations in Figure 1 are not sufficiently illustrative of the concepts explored in the paper nor do they do the author's work justice with regards to their experimental design.

- Missing NPLs in schematic for Figure 1
- Most experiments use NPLs for enzyme binding and assembly, but some do not. For example, experiments associated with Figure 5E and 5F appear to use 520 QDs, but the previous experiment in 5D used NPLs. The conclusions would seem to hold regardless, and thus it is not necessary to repeat these with NPLs, but it would clear up this matter to gain insight whether this was because the 520 QDs worked better here in these cases, or whether it was simply performed with these initially and not expanded to the other QDs and NPLs for testing.
- In the sequential addition experiments, are the enzyme amounts kept the same in the 1/7 QD additions. Does this mean every separate incubation has a different enzyme/QD ratio? This in reference to experiments associated with Figure 4D.
- Some recent studies related to the properties of enzymes on particles should be cited, for example, Xiong et al doi.org/10.1021/acsnano.2c00475

- Throughout the study, the authors talk about reaction kinetics, but if just the kinetics were different than it is unclear why the final concentrations appear to saturate at concentrations related to the kinetics. There doesn't seem to be a direct reason stated in the paper for these results, and this would need to be addressed.

The work elucidates the critical aspects of assembling enzymes, and it describes a useful and accessible methodology for enzymatic assembly. The work is potentially suitable for the publication in Nature Communications.

Reviewer #3 (Remarks to the Author):

The present work "Self-Assembled Nanoparticle Enzyme Clusters: An Emergent System for Accessing Substrate Channeling in Multienzymatic Cascades" reports enzyme-nanoparticle clusters, which can improve multi-step enzyme reactions (catalytic flux) by orders of magnitude, likely by driving substrate channeling. In their previous studies, the authors demonstrated enhanced enzymatic activities on nanoparticle surfaces, and more recently they immobilized two cascade enzymes on quantum dots (QDs), resulting in over 100-fold enzymatic activity improvement (ACS Nano 2018). It was also demonstrated that this two-enzyme cascade reaction was enhanced by intermediate channeling.

In the present work, the authors further extended to cascade reactions by 4 to 10 multi-saccharification or glycolytic enzymes. Enzyme-nanoparticle clusters were assembled following the same strategy that the authors reported (ACS Nano 2018). Clusters were assembled via interactions between QD surfaces and 6His tags of enzymes, where many enzymes are multimeric and thereby drive particle-enzyme clustering. Three CdSe/CdS/ZnS core/shell/shell QDs (520, 600, 660) and CdSe/ZnS core/shell nanoplatelets were used as clustering scaffolds. Cluster formation was analyzed by TEM and various other methods such as gel-based and DLS analyses. Nanoplates provided the most enzyme clustering and resulting the highest activity enhancement.

Overall, this study reports a simple but effective strategy to assemble enzyme clusters and to drive substrate channeling in multienzymatic cascade reactions up to 10 enzymes. All enzyme reactions were well characterized and clustered reactions were also well optimized. On the other hand, principles to assemble enzyme-QD clusters and channeling effects of these clusters were already reported by the authors in their previous study. And thereby, more complete establishment of this method is expected for publication of this work in Nature Communications. For example, an important remaining question would be if this method can be more generally used with other nanoparticle scaffolds. In addition, it would be valuable to compare QD-enzyme clusters to enzyme clusters that are formed by other methods.

Comments

1. As mentioned above, it is important to verify this method with other nano-structures. For example, gold nanoparticles can also interact with 6His tags. Can gold nanoparticles or plates with similar sizes (5-15 nm) co-cluster with enzymes and perform substrate channeling? Representative demonstration with 4 enzymes (or 7E) would greatly strengthen this work.
2. Similarly, can we compare the present NP-enzyme clusters to simple enzyme agglomerates (even random aggregates)? It would be essential to confirm the effect of enzyme immobilization on nano-structure surfaces for channeling, compared to simple agglomeration. For example, proteins can be aggregated (precipitated) by poly-phenols like tannic acids. Or simple addition of Ni²⁺ ions likely aggregates 6His-tagged enzyme mixtures, which contain multimeric proteins. Again, representative examination with 4 enzymes (or 7E) would be interesting.
3. Many supporting Figures (e.g. S2-S7...) are not referred in the text, and this makes reading the manuscript somewhat difficult.
4. A key conclusion of this work is 'larger cluster size is associated with better overall catalysis for QDs, and for NPLs up to a point, presumably by incorporating more enzymes at higher density'. Therefore, cluster size must be accurately determined. Here, cluster sizes of diverse systems were

mostly compared by TEM. DLS also provides important size distribution information for nano-structures. However, there is only one DLS data (Figure S31) for 520 QDs. I wonder if it is possible to obtain DLS data for other particles such as nanoplates (larger clusters) and 600/660 QDs for proper (better) size comparison.

Author responses to Reviewer Comments

REVIEWER COMMENTS

Reviewer #1 (Remarks to the Author):

Comment: Authors do not try to introduce immobilization. Modern enzyme immobilization, far to be only a way to ensure enzyme recovery and reuse, is nowadays a tool to improve many enzyme features, such as stability, activity, selectivity, specificity, inhibitions, even enzyme purification may be coupled. There are reviews in each of these topics, and at least this need to be mentioned. Because one of the problems of coimmobilization is that all enzyme are immobilized following the same strategy, that may be not optimal for all enzymes. Moreover, as reviewed by Arana, enzyme coimmobilization has many other problems. They cannot ignore this in introduction, advantages are not the only point to be pointed. Not in all cases the use of individually immobilized/optimized enzymes can be discarded in favor of coimmobilization.

Authors Response: The Reviewer seems to focus primarily on three issues in his remarks: namely that of enzyme immobilization chemistry, the papers we cite in our manuscript, and then what we interpret from their language as the issue of localized substrate sequestration effects, although the exact meaning of that point is hard to interpret. Based on these points, the Reviewer summarily rejects our manuscript and its findings *without even examining or discussing any of the experimental content*. As for the issue of immobilization, we agree with the Reviewer that our method of attaching the enzymes to the nanoparticles may not be optimal for all enzymes, but they should appreciate that within the context of the channeled systems we demonstrate: (i) it works; (ii) it is the same for all the enzymes; (iii) it gives rise to the nanoaggregate architecture that is so critical for accessing channeling; (iv) it is based on the simplest of bioconjugation chemistries – self-assembly and this is easily adoptable by others; (v) expandable to other materials as we show in the revised manuscript; (vi) in our system, we measure enzyme kinetic activity both on and off nanoparticle and critically use that to simulate the individual contributions and to improve channeled flux in the systems; and (vii) perhaps most importantly, it is non-covalent and non-electrostatic and so the enzyme activities are minimally perturbed in the aggregate structure. Covalent chemistries and other chemical modifications can significantly alter the activity of the enzymes and also of the system itself in a manner that will vary from batch to batch and would not allow for adequate simulations and optimization of the flux which are key to our process. We now emphasize these latter points in our revised manuscript. We also note that Reviewer 2 states the exact opposite of this Reviewer with their first statement “This study is quite in depth and highlights a relatively universal method for local enhancement of enzymatic cascades with commonly-used His-tag additions to proteins and readily available assembly materials.” We have also added the Arana citation on immobilization they mention to the revised manuscript (now ref. 46).

Comment: Authors make emphasis on enzyme proximity, when there are many reports that conclude that the main effect is the increment of intermedium concentration due to space limitations;

Authors Response: We are not sure what the statement “main effect is the increment of intermedium concentration due to space limitations” exactly means? Given that this Reviewer lists several of Hess’s papers below as examples, we interpret this as referring to the localized substrate-sequestration effects that are commonly misinterpreted as channeling, which is one of Hess’s main points in these references. Another major point of Professor Hess’s argument is that many of the studies utilize the glucose oxidase-horse radish peroxidase (Gox-HRP) as their enzyme system, which he and Wheeldon have shown as being in appropriate. Many of the papers the Reviewer lists below focus on this enzyme pair. All these points are discussed in detail in the manuscript and our exhaustive sets of controls, kinetic studies of individual enzymes, and numerous assay tests of channeling prove almost conclusively that what we see is channeling. Unfortunately, this point seems to be one of several that are lost on this reviewer. As to the specific citations this Reviewer mentions:

Zhang, Y., Tsitkov, S., Hess, H.

7601307821;57190762872;7201886138;

Proximity does not contribute to activity enhancement in the glucose oxidase-horseradish peroxidase cascade (2016) Nature Communications, 7, art. no. 13982, . Cited 182 times.
<https://www.scopus.com/inward/record.uri?eid=2-s2.0-85006982391&doi=10.1038/ncomms13982&partnerID=40&md5=46667cd79cf6c2c27f9735663564d1fb> DOI: 10.1038/ncomms13982

Authors Response: This paper was already cited and discussed in our manuscript and its primary point is that Gox-HRP are not appropriate for channeling studies, which we agree with and which is the reason we do not use those enzymes.

Xiong, Y., Tsitkov, S., Hess, H., Gang, O., Zhang, Y.

57208566514;57190762872;7201886138;6701826437;7601307821;

Microscale Colocalization of Cascade Enzymes Yields Activity Enhancement
ACS Nano, . In press DOI: 10.1021/acsnano.2c00475

Authors Response: This citation was not available when we wrote the original manuscript. We have added this citation to our manuscript. Similar to the comment above, we point out that this citation utilizes the Gox-HRP coupled enzyme pair, which are not appropriate for channeling studies.

Idan, O., Hess, H.

37124124000;7201886138;Origins of activity enhancement in enzyme cascades on scaffolds (2013) ACS Nano, 7 (10), pp. 8658-8665. Cited 92 times.

<https://www.scopus.com/inward/record.uri?eid=2-s2.0-84886995368&doi=10.1021/nn402823k&partnerID=40&md5=db9d6dc8e376931f6222d42f7cf6749b> DOI: 10.1021/nn402823k

Authors Response: This paper was already cited and discussed in our manuscript.

Idan, O., Hess, H. 37124124000;7201886138
Engineering enzymatic cascades on nanoscale scaffolds
(2013) Current Opinion in Biotechnology, 24 (4), pp. 606-611. Cited 57 times.
<https://www.scopus.com/inward/record.uri?eid=2-s2.0-84880960930&doi=10.1016/j.copbio.2013.01.003&partnerID=40&md5=99dd41dcd19ad3e1d096fc8a8bfe2664>DOI: 10.1016/j.copbio.2013.01.003

Authors Response: This paper was also already cited and discussed in our manuscript.

Idan, O., Hess, H.
37124124000;7201886138;
Diffusive transport phenomena in artificial enzyme cascades on scaffolds
(2012) Nature Nanotechnology, 7 (12), pp. 769-770. Cited 19 times.
<https://www.scopus.com/inward/record.uri?eid=2-s2.0-84874588554&doi=10.1038/nnano.2012.222&partnerID=40&md5=4aff3cb53ba41a916be9d01b6bbf1f3a>
DOI: 10.1038/nnano.2012.222
DOCUMENT TYPE: Letter

Authors Response: This paper was also already cited and discussed in our manuscript. The fact that 4 out of 5 of the papers this Reviewer mentions were already in the manuscript and used to cite the points they make and the 5th came out after the manuscript was completed strongly suggests that the Reviewer did not read our manuscript!

Comment: They cited some papers where the authors show that the positive effects are due to other effects different to substrate channeling. However, in these reports, if the enzyme proximity is so important, this should be also a main factor in that papers.

Authors Response: This is another point that is very hard to interpret linguistically and without specifics on which cited papers the Reviewer is referring to. We counter that in addition to the current exhaustive study which we are submitting here, three of our own papers cited in the manuscript where channeling is indeed shown in nanoparticle-enzyme aggregates *albeit* in simpler enzymatic systems (2-coupled enzymes, Pyk-LDH ref 55 and Bal-RADH ref. 61; 3-coupled enzymes Amylase, Maltase, and Glucokinase ref. 43) --- enzyme proximity via coassembly into a nanoparticle aggregate is a critical determinant of access to channeling. Further, when examining the same 3-coupled enzyme system as assembled on DNA at much lower ratios, we point out that each enzyme displays enhanced activity but no concerted channeling (ref .51). This latter result we ascribe to the same enhancing effects that Wheeldon and Hess describe for DNA assembled enzymes and we suggest that the lack of channeling in this system has to do with not achieving the requisite density of enzymes in close proximity to allow for channeling. Again, these points are all discussed in the manuscript and seem all to be lost on this reviewer as they only espouse their own viewpoint without examining or discussing any of the extensive experimental data presented in the manuscript.

Comment: If they want to make a deeper study on this, the paper could be a very good one, To ensure that the substrate channeling is the key, they should combine sequentially immobilized

enzymes (in form of crowns, where only the enzyme in the border of the crown are bear to the other sequential enzymes) using “logic and illogic” enzyme sequences, and enzyme colocalization, to confirm that the effect of the substrate channeling and not the most evident increase of substrate concentration for each enzyme.

Authors Response: Again, without examining or discussing our extensive experimental data, this Reviewer now suggests that we undertake a completely different experimental system to prove our point and make the study outcome acceptable to them. We have no idea what the ‘crowns’ are, nor ‘where only the enzyme in the border of the crown are bear to the other sequential enzymes)’ along with what “logic and illogic” enzyme sequences are? If the crowns are crown ethers, then that is a completely different effect due to addition of an organic molecule to which only certain classes of enzymes can tolerate to and is being misconstrued here.

Comment: In any case, the current paper seems to be a new step on a previously described strategy, including more enzymes. In my opinion, this not justify the publication in a highly repute djournl such as Nature communications.

Authors Response: We respectfully but very strongly disagree with the conclusion from this Reviewer based on all the points iterated above and also that of the other Reviewers which actually focus on our experimental data. Moreover, after having denied that we achieve channeling throughout their entire review up to this point – if their statement here is taken at face value “the current paper seems to be a new step on a previously described strategy, including more enzymes” then they, in essence, concede that we did indeed achieve channeling all along because that is what those papers are unambiguously about and this counters all their previous points. Overall, this is another confusing point in a review that is very hard to understand and interpret let alone respond to!

Reviewer #2 (Remarks to the Author):

Comment: This study is quite in depth and highlights a relatively universal method for local enhancement of enzymatic cascades with commonly-used His-tag additions to proteins and readily available assembly materials. As the authors have stated, this work is mean to explore and clarify concepts that have been both theoretically and experimentally explored using different systems. Moreover, many systems have used particles/surfaces for assembly as well. The presented experimental design elucidates that as opposed to directly scaffolded systems with particular attention paid to spacing and enzyme order, there is significant, impactful benefit to more broadly-considered enzyme ordering and packing. The authors have expanded the enzyme systems explored in relation to other works by considering a larger and more complex cascade. Overall, this is an interesting, timely and important work which will be of interest to a broad community, thus, in my opinion, it might be suitable for NatureCommunications. Several comments should be addressed, as listed below

Authors Response: The Authors thank the Reviewer for their diligent reading of the manuscript. We recognize that in addressing their comments, the manuscript has clearly been significantly improved.

Comment: The writing of the paper is quite dense, and figures beyond Figure 1 would benefit from at least minimal schematics, as each is tied together conceptually but doesn't have illustrative aspects defining the core of this paper, which is assembly considerations. The aggregations in Figure 1 are not sufficiently illustrative of the concepts explored in the paper nor do they do they author's work justice with regards to their experimental design.

Authors Response: We appreciate the Reviewers point here. To help improve data presentation, we have added an indication into each plot of which exact enzyme system is being examined there. We have also added a conceptual schematic in the middle right of Figure 1A to depict how the multienzymatic channeling overcomes diffusion limitations and have added the following text to the Figure 1A legend: "Forming into NP-enzyme clusters and engaging in multistep channeling increases the overall catalytic flux by orders of magnitude over that of freely diffusing enzymes, which encounter significant diffusion limitations. The latter substantially reduces the overall transient time (τ) for that reaction."

- Missing NPLs in schematic for Figure 1

Authors Response: We have added the NPLs to the schematic in Figure 1.

- Most experiments use NPLs for enzyme binding and assembly, but some do not. For example, experiments associated with Figure 5E and 5F appear to use 520 QDs, but the previous experiment in 5D used NPLs. The conclusions would seem to hold regardless, and thus it is not necessary to repeat these with NPLs, but it would clear up this matter to gain insight whether this was because the 520 QDs worked better here in these cases, or whether it was simply performed with these initially and not expanded to the other QDs and NPLs for testing.

Authors Response: We thank the reviewer for bring up this point and have now clarified this by adding the following sentence "Subsequent experiments utilized either NPLs or 520 QDs somewhat interchangeably as they manifest the best increases in catalytic flux." in the section entitled "Initial Assays, Optimization of Enzyme Ratios, and Assay Conditions." after the description of testing different QD sizes and NPLs.

- In the sequential addition experiments, are the enzyme amounts kept the same in the 1/7 QD additions. Does this mean every separate incubation has a different enzyme/QD ratio? This in reference to experiments associated with Figure 4D.

Authors Response: We have answered this question by adding the following text to the Figure 4D legend "NPLs assembled with 7E cascade at Opt 2 ratios in batch (enzymes mixed together first followed by NP addition) or separately where each enzyme was added to 1/7th the amount of NPL at the same concentration and ratio present in batch and then combined together just prior to the start of the assay as designated by 'separate assembly'."

- Some recent studies related to the properties of enzymes on particles should be cited, for example, Xiong et al doi.org/10.1021/acsnano.2c00475

Authors Response: This paper has now been added to the revised manuscript as a citation.

- Throughout the study, the authors talk about reaction kinetics, but if just the kinetics were different than it is unclear why the final concentrations appear to saturate at concentrations related to the kinetics. There doesn't seem to be a direct reason stated in the paper for these results, and this would need to be addressed.

Authors Response: The Reviewer brings up an excellent point that we overlooked and did not explain in the first version of the manuscript. In the section entitled 'Initial Assays, Optimization of Enzyme Ratios, and Assay Conditions.' We have now added the following sentence to clarify this issue "In examining the enzyme assay plots in **Figure 2** and those described below, optimized channeling sometimes increases the kinetic flux to the point that apparent saturation is reached in the assay's time window. It is important to note that in all the other configurations plotted along with this type of data, the trajectories indicate a similar rise to saturation *albeit* at a much slower rate as expected. We also do not account for any effects from the reverse gluconeogenic reactions, which will be initiated when a high enough concentration of a given intermediary is reached and which may slow down flux and final product formation."

Comment: The work elucidates the critical aspects of assembling enzymes, and it describes a useful and accessible methodology for enzymatic assembly. The work is potentially suitable for the publication in Nature Communications.

Authors Response: We again thank the Reviewer for their diligent reading and helpful comments.

Reviewer #3 (Remarks to the Author):

Comment: The present work "Self-Assembled Nanoparticle Enzyme Clusters: An Emergent System for Accessing Substrate Channeling in Multienzymatic Cascades" reports enzyme-nanoparticle clusters, which can improve multi-step enzyme reactions (catalytic flux) by orders of magnitude, likely by driving substrate channeling. In their previous studies, the authors demonstrated enhanced enzymatic activities on nanoparticle surfaces, and more recently they immobilized two cascade enzymes on quantum dots (QDs), resulting in over 100-fold enzymatic activity improvement (ACS Nano 2018). It was also demonstrated that this two-enzyme cascade reaction was enhanced by intermediate channeling.

In the present work, the authors further extended to cascade reactions by 4 to 10 multi saccharification or glycolytic enzymes. Enzyme-nanoparticle clusters were assembled following the same strategy that the authors reported (ACS Nano 2018). Clusters were assembled via interactions between QD surfaces and 6His tags of enzymes, where many enzymes are multimeric and thereby drive particle-enzyme clustering. Three CdSe/CdS/ZnS core/shell/shell QDs (520, 600, 660) and CdSe/ZnS core/shell nanoplatelets were used as clustering scaffolds. Cluster formation was analyzed by TEM and various other methods such as gel-based and DLS analyses. Nanoplates provided the most enzyme clustering and resulting the highest activity enhancement.

Overall, this study reports a simple but effective strategy to assemble enzyme clusters and to drive substrate channeling in multienzymatic cascade reactions up to 10 enzymes. All enzyme reactions

were well characterized and clustered reactions were also well optimized. On the other hand, principles to assemble enzyme-QD clusters and channeling effects of these clusters were already reported by the authors in their previous study. And thereby, more complete establishment of this method is expected for publication of this work in Nature Communications. For example, an important remaining question would be if this method can be more generally used with other nanoparticle scaffolds. In addition, it would be valuable to compare QD-enzyme clusters to enzyme clusters that are formed by other methods.

Authors Response: The Authors thank this Reviewer for their diligent reading of the manuscript. We recognize that in addressing their comments, the manuscript has clearly been significantly improved.

Comments

1. As mentioned above, it is important to verify this method with other nano-structures. For example, gold nanoparticles can also interact with 6His tags. Can gold nanoparticles or plates with similar sizes (5-15 nm) co-cluster with enzymes and perform substrate channeling? Representative demonstration with 4 enzymes (or 7E) would greatly strengthen this work.

Authors Response: The Reviewer brings up an excellent point here, which we felt helped strengthen the manuscript significantly through addressing. Towards ‘other nano-structures’ we undertook a significant expansion and revision of our work and now demonstrate enhanced kinetic flux in the 7E system with three other nanomaterials used as both scaffold and enzyme crosslinker. We utilize 5 nm diameter gold nanoparticles, commercial QD materials, and, as opposed to the two previous ‘hard’ nanoparticle materials, a third softer dendrimer. This is now presented in detail in the new section added to the manuscript entitled “Channeling in Other Self-Assembled Nanomaterial-Enzyme Clusters” along with the newly added Figure 6 . We note in the revised manuscript that gold nanoparticles with the requisite surface chemistry to accomplish the same type of experiments are available from several vendors as is the dendrimer. Importantly, all of the new materials we have now tested are commercially available and should allow others to pursue similar formats without the need for our in-house synthesized QD and NPLs.

2. Similarly, can we compare the present NP-enzyme clusters to simple enzyme agglomerates (even random aggregates)? It would be essential to confirm the effect of enzyme immobilization on nano-structure surfaces for channeling, compared to simple agglomeration. For example, proteins can be aggregated (precipitated) by poly-phenols like tannic acids. Or simple addition of Ni²⁺ ions likely aggregates 6His-tagged enzyme mixtures, which contain multimeric proteins. Again, representative examination with 4 enzymes (or 7E) would be interesting.

Authors Response: The Reviewer brings up another interesting point here. This one, however, is much harder to address. One of the primary precepts in our manuscript and our process for accessing channeling with nanomaterial-enzyme aggregates if you will (including the 3 new materials we added in point 1 above) is that all the enzymes are immobilized to the nanoparticles using the exact same chemistry (His₆-based metal affinity) and this also drives subsequent aggregate formation. Equally importantly, because this assembly does not perturb the enzyme activity and, in many cases, enhances the enzyme activity in a manner we can measure – we are

able to use numerical simulations to simulate each enzyme's individual contribution in a model channeled system and to predict optimized enzyme ratios to utilize in our experiments.

The Reviewer suggests two ways of forming enzyme only agglomerates, using tannic acids and excess Ni²⁺ ions. The other more common way found in the literature is with the protein crosslinker agent glutaraldehyde. Given the importance of this point we undertook testing of them towards creating protein only agglomerates. Initial assays with each of these methods show that they each will significantly reduce enzyme activity especially in the 7E system (when free in solution) in a manner that is unpredictable and which we confirmed varied between reactions and from batch to batch. This negates our ability to numerically simulate the contribution of each enzyme within a full cascade and model the systems in order to optimize the flux. If implemented, this approach would also now require a cleanup step after preparation as opposed to our 'mix-and-go' strategy which we believe is a major asset. Moreover, that the systems now perform worse than even the free enzyme systems (and not even comparing to the nanoparticle-assembled systems) resulted in us not pursuing this further. We now mention this in the revised manuscript in the newly added section and present some representative data in the revised SI section.

Directly related to this, we had assumed that the difference between our systems and those assembled using chemical approaches to create enzyme agglomerates or cross-linked enzyme aggregates (CLEA's) was implicit and did not explain this point in the first version of the manuscript. We have now added text in the conclusions, which explicitly addresses this latter point with the following in the Discussion and Conclusions section:

“One of the key design precepts behind the assemblies described here, and that we argue is critical for both function and accessing channeling, is to minimize any chemical modifications or assembly strategies that deleteriously perturb native enzyme function. The highest catalytic rate of each participant enzyme is required and the ability to achieve this in a robust and predictable manner is also critical to accurately modeling the contribution of that enzyme within a (channeled) cascade using numerical simulations. For these reasons we regard our exploitation of His₆ metal-affinity coordination as a key enabler. Importantly, such coordination relies on residues located at each enzymes's distal termini, which, in turn, allows all enzymes to assemble in the same non-perturbative manner, providing for increased enzyme activity in many cases, and allowing us to undertake detailed modeling of each enzyme's channeled contribution to its given system's cascade in order to improve flux. In contrast, cross-linked enzyme aggregates (CLEAs) have been suggested as an alternative means of creating similar enzyme clusters without requiring an additional nanoscaffold.^{90, 91} Although extremely useful in many applications, it is well recognized that CLEAs commonly display reduced enzymatic activity due to the chemistry utilized in their assembly.^{90, 91} To ascertain if CLEAs could potentially function in a similar manner to our clusters, we selected tannic acid and Ni²⁺ as two non-covalent assembly mechanisms along with glutaraldehyde as a third covalent chemistry for testing.⁹²⁻¹⁰² Initial assays (see **Figures S45-S47**) confirmed that each of these agents did indeed deleteriously alter activity in the 7E system. Moreover, they modified the activity of individual enzymes in an unpredictable manner that varied between batch-to-batch preparations (data not shown). Indeed, the CLEA literature itself also points out that most methods of forming such structures will crosslink the enzymes in an unpredictable manner and a decrease in enzyme activity is to be expected.⁹⁰⁻⁹⁴ Given the reductions in activity along with the issues of unpredictability (*viz-à-viz* performing numerical simulations), we did not pursue this

system further. Reliance on His₆ metal-affinity coordination is not however limiting and this was confirmed by our demonstrating that clustered assembly of enzymes and access to channeling are achievable in three other NP systems, namely, AuNPs, dendrimers, and a commercially available QD with a different surface chemistry.”

Overall, the main point to be appreciated about this is that due to their assembly chemistry CLEAs and similarly assembled enzyme aggregates are very different structurally and functionally and so they are not compared to our systems further. Indeed, the CLEA literature itself also points out that most methods of forming such structures will crosslink the enzymes in an unpredictable manner and a decrease in enzyme activity is to be expected. If CLEAs functioned well for achieving channeling in enzyme cascades, they would already have been well-established for these purposes given the strong interest in channeling.

3. Many supporting Figures (e.g. S2-S7...) are not referred in the text, and this makes reading the manuscript somewhat difficult.

Authors Response: In the revised manuscript, we have tried to increase the number of callouts to data in the SI where appropriate. However, the amount of content in the SI does not allow us to refer to everything.

4. A key conclusion of this work is ‘larger cluster size is associated with better overall catalysis for QDs, and for NPLs up to a point, presumably by incorporating more enzymes at higher density’. Therefore, cluster size must be accurately determined. Here, cluster sizes of diverse systems were mostly compared by TEM. DLS also provides important size distribution information for nano-structures. However, there is only one DLS data (Figure S31) for 520 QDs. I wonder if it is possible to obtain DLS data for other particles such as nanoplates (larger clusters) and 600/660 QDs for proper (better) size comparison.

Authors Response: The Reviewer brings up an important point that we struggled with ourselves when we initially considered using DLS analysis to characterize our clusters. The DLS data we present in Figure S34 shows the overall difference in size between samples where each sample is made up by an underlying distribution of clusters with different sizes. In essence – this figure shows from a very simplistic perspective – here is how the size distribution changes between these samples. As our in house DLS expert pointed out to us, the primary issue with applying DLS analysis to extract out the underlying range of sizes of the mixed cluster samples we utilize is that signal intensity of larger structures scales or increases by the 6th power! This means that a single 50 nm diameter cluster will have the same signal intensity as one million 5 nm clusters! (see for example the following references - [https://chem.libretexts.org/Bookshelves/Analytical_Chemistry/Physical_Methods_in_Chemistry_and_Nano_Science_\(Barron\)/02%3A_Physical_and_Thermal_Analysis/2.04%3A_Dynamic_Light_Scattering](https://chem.libretexts.org/Bookshelves/Analytical_Chemistry/Physical_Methods_in_Chemistry_and_Nano_Science_(Barron)/02%3A_Physical_and_Thermal_Analysis/2.04%3A_Dynamic_Light_Scattering); see also *Supramol Chem.* 2019 ; 31(9): 608–615. doi:10.1080/10610278.2019.1629438.) Taking this into consideration in the context of our samples which have a wide dispersion of different sized clusters within them reveals the complexity of the issue and why we could not usefully apply DLS analysis. How do you extract out true size information versus high intensity? The one time we did apply it in the SI was to just show a change in average size range where this is not a major issue . Thus, we turned to TEM

analysis to at least gather some semi-quantitative supporting information even though this was far more arduous and time consuming.

REVIEWERS' COMMENTS

Reviewer #2 (Remarks to the Author):

My comments were fully addressed in the reply and the related changes were incorporated in the revised manuscript. I recommend the paper for the publication.

Reviewer #3 (Remarks to the Author):

The authors properly addressed all raised issues in their revised manuscript. In particular, testing other nano-structure scaffolds nicely validates the universal applicability of this work. Therefore, now I recommend publication of this work.